# SLiM : One-shot Quantized Sparse Plus Low-rank Approximation of LLMs

## Abstract

Large Language Models (LLMs) have revolutionized natural language understanding and generation tasks but suffer from high memory consumption and slow inference times due to their large parameter sizes. Traditional model compression techniques, such as quantization and pruning, mitigate these issues but often require retraining to maintain accuracy, which is computationally expensive. This paper introduces SLiM , a novel approach for compressing LLMs using a one-shot Quantized Sparse Plus Low-rank Approximation. SLiM eliminates the need for costly retraining by combining a symmetric quantization method (SLiM-Quant ) with a saliency-based low-rank approximation. Our method reduces quantization error while leveraging sparse representations compatible with accelerated hardware architectures. Additionally, we propose a parameter-efficient fine-tuning recipe that significantly reduces overhead compared to conventional quantization-aware training. SLiM achieves up to a 5.4% improvement in model accuracy for sparsity patterns like 2:4, and the fine-tuning step further enhances accuracy by up to 5.8%, demonstrating state-of-the-art performance. This work provides a pathway for efficiently deploying large models in memory-constrained environments without compromising accuracy.[1]

## 1 Introduction

Large Language Models (LLMs) (Brown et al., 2020; Radford et al., 2019) are transformative for natural language understanding and generation (Suzgun et al., 2022; Zhou et al., 2023); however, their extensive parameter count leads to large memory footprints and longer inference times, making them expensive to execute. Model compression methods, such as sparsity and quantization, have shown promising results in reducing the inference costs of LLMs. However, these methods often require an expensive retraining procedure on large amounts of data to restore the original model accuracy (Sanh et al., 2020; Park et al., 2018), while facing numerical and optimization stability challenges when dealing with quantized weights in quantization-aware-training (Gholami et al., 2022).

To address these issues, one-shot pruning methods have emerged, eliminating the need for the retraining and achieve high accuracy using only a small set of calibration data. Optimal Brain Damage (OBD) (LeCun et al., 1989) pioneered the use of second-order information of the loss function for model compression (Singh & Alistarh, 2020; Mozaffari et al., 2023), though at a high computational cost. Subsequent methods like Optimal Brain Surgeon (OBS) (Hassibi et al., 1993) and modern approaches such as SparseGPT (Frantar & Alistarh, 2023) and WANDA (Sun et al., 2023) build on these ideas, introducing computationally feasible alternatives for LLMs. While these methods perform well with unstructured sparsity, they struggle with semi-structured sparsity patterns like the NVIDIA 2:4 sparsity pattern (Mishra et al., 2021), which are necessary for hardware-accelerated inference.

To address these issues, one-shot pruning methods have emerged, eliminating the need for the retraining and achieve high accuracy using only a small set of calibration data. Optimal Brain Damage (OBD) (LeCun et al., 1989) pioneered the use of second-order information of the loss function for model compression (Singh & Alistarh, 2020; Mozaffari et al., 2023), though at a high computational

---

[1]Code and data for SLiM is available at: `https://anonymous.4open.science/r/slim`

cost. Subsequent methods like Optimal Brain Surgeon (OBS) (Hassibi et al., 1993) and modern approaches such as SparseGPT (Frantar & Alistarh, 2023) and WANDA (Sun et al., 2023) build on these ideas, introducing computationally feasible alternatives for LLMs. While these methods perform well with unstructured sparsity, they struggle with semi-structured sparsity patterns like the NVIDIA 2:4 sparsity pattern (Mishra et al., 2021), which are necessary for hardware-accelerated inference.

While sparsity and quantization individually offer substantial reductions in model size and inference cost, combining these techniques holds even greater potential for compressing large models (Frantar & Alistarh, 2023). However, combining sparsity and quantization often exacerbates the accuracy loss from each method, resulting in a substantial performance gap between compressed and original models. This accuracy gap highlights the need for further innovations in compression techniques. Recent work has aimed to reduce compression error by using learnable low-rank adapters to minimize weight reconstruction error (Guo et al., 2023; Nikdan et al., 2024), followed by an expensive fine-tuning step on hundreds of millions of tokens (Dettmers et al., 2023; Li et al., 2023). This prolonged fine-tuning is necessary because the low-rank adapters do not account for the saliency of the weights at initialization (Dettmers et al., 2023; Guo et al., 2023).

To address these limitations, we propose a saliency-based one-shot low-rank approximation that mitigates the accuracy loss caused by quantization and sparsity. Saliency-based methods require weights to remain static during quantization and pruning, which renders approaches like OPTQ and SparseGPT ineffective. To resolve this, we introduce a novel symmetric weight quantization scheme that not only effectively reduces quantization error but is also compatible with saliency-based methods. We complement our quantizer and low-rank adapters with an optional light-weight fine-tuning recipe that can further boost the accuracy of the models using only 300,000 tokens.

Our method, SLIM , is a One-shot Quantized **S**parse Plus **L**ow-rank Approx**i**mation of LL**M**s. Key contributions of SLIM are:

- **SLIM-Quant**: We propose a symmetric weight quantization scheme that minimizes the Frobenius norm of quantization error without altering the weights, making it compatible with saliency-based pruning and low-rank approximation methods. Unlike group quantization, SLIM-Quant uses a single parameter for the entire weight matrix, reducing computational and memory overhead and simplifying implementation.

- **Saliency-based One-shot Low-rank Adapters**: We introduce a one-shot low-rank adapter method that minimizes accuracy loss by reconstructing weights based on their saliency, targeting weights with the highest impact on model output.

- **Parameter-Efficient Fine-tuning**: We propose a fine-tuning recipe for sparse, quantized models that avoids the complexities of quantization-aware training while drastically reducing fine-tuning time. For example, fine-tuning a 13B parameter model, which typically takes up to 36 days, is reduced to just 14 hours on a single H100 GPU.

- **Accuracy Gains**: SLIM improves model accuracy by up to 5.4% (LLaMA-2-7B) over state-of-the-art pruning and quantization methods (SparseGPT + Group OPTQ) for 2:4 sparsity. With our parameter-efficient fine-tuning, the gap widens to 5.8% (LLaMA-2-13B).

## 2 PRELIMINARIES

**Optimal Brain Surgeon.** The primary objective in model compression is to minimize the output discrepancy between the compressed and original models. Optimal Brain Surgeon (Hassibi et al., 1993) simplifies this approach by minimizing the output difference at each network layer over a calibration dataset. Consider a single feed-forward layer with input $\mathcal{X} \in \mathbb{R}^{b \times d_{in}}$, weight $\mathcal{W} \in \mathbb{R}^{d_{in} \times d_{out}}$, and output $\mathcal{Y} \in \mathbb{R}^{b \times d_{out}}$, where $b$, $d_{in}$, and $d_{out}$ represent the batch size, input hidden dimension, and output hidden dimension, respectively. Denoting the compressed matrices with a superscript $C$, OBS aims to minimize Equation 1. This formulation allows OBS to focus on maintaining layer-wise fidelity during compression, potentially leading to better overall model performance.

$$\min_{\mathcal{W}^C} |\mathcal{Y}^C - \mathcal{Y}|^2 = \min_{\mathcal{W}^C} |\mathcal{X}(\mathcal{W} - \mathcal{W}^C)|^2 \tag{1}$$

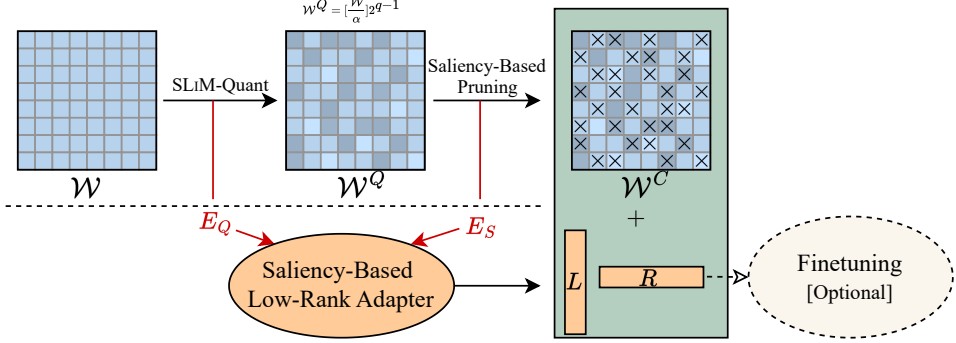

Figure 1: Figure 1: SLɪM weight compression pipeline. The process involves: (1) Quantizing weights using the symmetric SLɪM-Quant algorithm (generating quantized weight $\mathcal{W}^Q$ and quantization error $E_Q$); (2) Sparsifying quantized weights through saliency-based pruning (generating compressed weight $\mathcal{W}^C$ with sparsity error $E_S$); (3) Compensating for quantization and sparsity errors via SLɪM saliency-based low-rank approximation (generating left and right low-rank adapters $L$ and $R$). Optionally, adapters can undergo fine-tuning to further improve model accuracy while keeping sparse quantized weights frozen.

Considering Equation 1, Wanda (Sun et al., 2023) proposes a saliency matrix $\mathcal{S}$ to evaluate the importance of each parameter. This matrix is defined as $\mathcal{S} = \mathbf{x}^T \odot \mathcal{W}$, where $\mathbf{x} \in \mathbb{R}^{d_{in} \times 1}$ represents the average absolute value of each input, $\odot$ denotes element-wise multiplication, and $|.|$ represents the absolute value operator. SLɪM adopts this same metric to assess the importance of different weights and determine the sparsity patterns in the sparse weight representation.

**Symmetric Quantization.** Symmetric quantization, the simplest form of quantization, computes $\mathcal{M}^Q \propto round(\frac{\mathcal{M}}{\alpha})$, where $\mathcal{M}^Q$ is the quantized matrix and $\alpha$ is the quantization parameter. This method effectively accelerates matrix multiplications, as dequantization of the output requires only a scalar multiplication.

AbsMax (Jacob et al., 2018), the most common symmetric quantization method, sets $\alpha$ to the maximum absolute value in the matrix. However, AbsMax is sensitive to outliers, which can significantly alter $\alpha$, thereby reducing quantization precision. For zero-centered bell-curved distributions typical of LLM weights and inputs, AbsMax maps a large portion of weights to zero, resulting in high error values.

Group quantization Alistarh et al. (2017); Gunho et al. (2022) addresses this issue by sharing the quantization parameter among fewer elements in the weight matrix. This approach divides the weight matrix into smaller subgroups, each quantized with its own scaling factor. While this allows for more fine-grained representation of weight distributions and better captures local variations in weight magnitudes, it introduces memory overheads and complicates dequantization methods. Specifically, group quantization increases memory usage and computational complexity in exchange for potentially improved quantization accuracy.

## 3 QUANTIZED SPARSE PLUS LOW-RANK APPROXIMATION OF LLMS

To effectively apply quantization, sparsity, and low-rank adapters to LLMs, SLɪM introduces a novel quantization scheme called SLɪM-Quant . This method reduces quantization error and is followed by pruning the quantized model using the importance metric proposed in Section 2. Subsequently, SLɪM adds low-rank adapters to minimize the saliency of the compression error introduced by sparsity and quantization. Figure 1 illustrates this process. In the following subsections, we discuss each step of this approach in detail.

### 3.1 SLiM Quantization Method

SLiM focuses on symmetric weight quantization due to its low dequantization and memory overhead and ease of implementation. Denoting the quantized matrices by $Q$ superscript, Equation 2 shows the symmetric quantization formula for $q$-bit quantization, where $\alpha$ is the quantization parameter and $clip(.)$ operator clips the input to values between $[-1, 1]$.

$$\mathcal{W}^Q = round(clip(\frac{\mathcal{W}}{\alpha}))2^{q-1} \tag{2}$$

The objective of quantization is to reduce the weight reconstruction error shown in Equation 3, where the $*$ superscript shows the optimal value. But the objective function in Equation 3 is not convex, and to our best knowledge, does not have a closed form solution.

$$\alpha^* = \arg\min_{\alpha} ||\mathcal{W}^Q - \mathcal{W}||^2 = \arg\min_{\alpha} ||round(clip(\frac{\mathcal{W}}{\alpha}))2^{q-1} - \mathcal{W}||^2 \tag{3}$$

To solve the mean squared error (MSE) problem in Equation 3, we propose a probabilistic reformulation as shown in Equation 4, where $Q(.)$ and $Q^{-1}(.)$ are the quantization and dequantization functions respectively and $f(.)$ is the probability distribution function (PDF) of the weight elements.

$$\alpha^* = \arg\min_{\alpha} ||\mathcal{W}^Q - \mathcal{W}||^2 = \arg\min_{\alpha} \int_{-\infty}^{\infty} f(x)|Q^{-1}(Q(x)) - x|^2 dx \tag{4}$$

By incorporating the quantization formula from Equation 2 into Equation 4, we can simplify the integration into the sum of two terms based on the absolute value of the data: the quantization error for absolute values less than $\alpha$ (Equation 5) and the clipping error for absolute values larger than $\alpha$ (Equation 6). Here, $f_abs(.)$ represents the probability density function (PDF) of the absolute value of the weights. Equation 7 presents the simplified version of Equation 4.

$$E_{quant}(\alpha) = \int_0^{\alpha} f_{abs}(x)|\alpha \times round(\frac{x}{\alpha}) \times 2^{1-q} - x|^2 dx \tag{5}$$

$$E_{clip}(\alpha) = \int_{\alpha}^{\infty} f_{abs}(x)|\alpha - x|^2 dx \tag{6}$$

$$\alpha^* = \arg\min_{\alpha} E_Q(\alpha) = \arg\min_{\alpha} E_{quant}(\alpha) + E_{clip}(\alpha) \tag{7}$$

Equation 7 can theoretically be solved by differentiating the objective function with respect to $\alpha$, assuming the probability density function (PDF) of the distribution is known. However, in practice, neural network weight distributions don't follow standard distributions. We tested models like Gaussian, Laplace, Pareto, q-Gaussian, and Weibull, but none fit the observed weight distributions, highlighting their uniqueness.

To overcome the lack of a closed-form weight PDF, we use numerical integration over the weight histogram to solve Equation 7. We implement a multi-grid approach to optimize efficiency, starting with 10 uniform samples in the range $(0, \max(W))$ and iteratively refining around the minimum error to find the optimal $\alpha$. The full method is detailed in Algorithm 1.

### 3.2 SLiM Low-rank Adapters

The effects of quantization and pruning of a weight matrix can be modeled as additive noise, such that $\mathcal{W}^C = \mathcal{W} + E_Q + E_S$, where $E_Q = \mathcal{W} - \mathcal{W}^Q$ and $E_S = \mathcal{W}^C - \mathcal{W}^Q$ are the quantization and sparsity errors respectively. We aim to add low-rank adapters to the weights that cancel the compression errors, i.e. $\mathcal{W} \approx \mathcal{W}^C + \mathcal{L}\mathcal{R}$, where $\mathcal{L} \in \mathbb{R}^{d_{in} \times r}$ and $\mathcal{R} \in \mathbb{R}^{r \times d_{out}}$ are the low-rank adapters and $r$ is the adapter rank.

---

**Algorithm 1** SLIM-Quant Algorithm

---

1: **Input:** Weight Magnitude PDF: $f_{abs}$, High Resolution Step Size: $\eta_{high}$, Low Resolution Step Size: $\eta_{low}$ Weight Matrix: $\mathcal{W}$, Quantization Bitwidth: $q$
2: **Output:** $\mathcal{W}_{quant}$
3: **procedure** ESTIMATEERROR($\alpha$)
4: $\quad E_{quant}(\alpha) = \int_0^\alpha f_{abs}(x)|\alpha \times round(\frac{x}{\alpha}) \times 2^{1-q} - x|^2 dx$
5: $\quad E_{clip}(\alpha) = \int_\alpha^\infty f_{abs}(x)|\alpha - x|^2 dx$
6: $\quad$ **return** $E_{quant} + E_{clip}$
7: $E$ = EmptyDictionary()
8: **for** $\alpha$ in range(0, $M$, $\eta_{low}$) **do**
9: $\quad E(\alpha) = $ ESTIMATEERROR($\alpha$)
10: $\alpha_{low} = \arg\min_\alpha E(\alpha)$
11: **for** $\alpha$ in range($\alpha_{low} - \eta_{low}$, $\alpha_{low} + \eta_{low}$, $\eta_{high}$) **do**
12: $\quad E(\alpha) = $ ESTIMATEERROR($\alpha$)
13: $\alpha^* = \arg\min_\alpha E(\alpha)$
14: $\mathcal{W}_{quant} = round(clip(\frac{W}{\alpha^*})) \times 2^{q-1}$

---

In a naive attempt, one can try to minimize the total error norm between $\mathcal{W}$ and $\mathcal{W}^C$. In this approach, the magnitude of the error is the minimization objective, and the importance of different weights are not taken into account. Similar to magnitude pruning, this approach results in low accuracy in the model as discussed in Section 4.

We propose a new formulation to incorporate the weight saliency into the low-rank approximation, and then utilize the proper saliency function to find the optimal adapters in practice. Assuming that there exists an additive invertible saliency function $F : \mathbb{R}^{d_{in} \times d_{out}} \to \mathbb{R}^{d_{in} \times d_{out}}$, solving Equation 8, in which we have used the additive property of $F$ and the fact that $F(\mathcal{W}) = F(\mathcal{W}^C + (LR))$.

$$\mathcal{L}, \mathcal{R} = \arg\max_{\mathcal{L},\mathcal{R}} ||F(\mathcal{W}^C + \mathcal{L}\mathcal{R})||^2 = \arg\min_{\mathcal{L},\mathcal{R}} ||F(\mathcal{W} - (\mathcal{W}^C + \mathcal{L}\mathcal{R}))||^2 \qquad (8)$$

By using the additive feature of the saliency function $F(.)$, we can simplify Equation 8 to Equation 9. By solving the optimization problem in equation 9, we can obtain the optimal low-rank adapters.

$$\mathcal{L}, \mathcal{R} = \arg\min_{\mathcal{L},\mathcal{R}} ||F(\mathcal{W} - \mathcal{W}^C) - F(\mathcal{L}\mathcal{R})||^2 = \arg\min_{\mathcal{L},\mathcal{R}} ||F(-(E_Q + E_S)) - F(\mathcal{L}\mathcal{R})||^2 \qquad (9)$$

For solving the optimization problem in Equation 9, we replace $F(.)$ with the saliency function discussed in section 2, i.e. $F(\mathcal{W}) = \mathbf{x}^T \odot \mathcal{W}$, where $\mathbf{x} \in \mathbb{R}^{d_{in} \times 1}$ represents the average absolute value of inputs from a calibration set. For simplicity, we use matrix multiplications instead of element-wise multiplication, resulting in Equation 10.

$$F(\mathcal{W}) = diag(\mathbf{x})\mathcal{W} \qquad (10)$$

Replacing Equation 10 in the objective function in Equation 9, one can obtain Equation 11, which can be solved by a singular value decomposition and a matrix inversion on the left low-rank adapter, as in Equation 12.

$$\mathcal{L}, \mathcal{R} = \arg\min_{\mathcal{L},\mathcal{R}} || - diag(\mathbf{x})(E_Q + E_S) - diag(\mathbf{x})\mathcal{L}\mathcal{R}||^2 \qquad (11)$$

$$diag(\mathbf{x})\mathcal{L}, \mathcal{R} = -SVD(diag(\mathbf{x})(E_Q + E_S)) \qquad (12)$$

Please note that the invertibility of the saliency function in Equation 10 is dependent on the invertibility of $diag(\mathbf{x})$, which in turn requires all values in $\mathbf{x}$ to be not equal to zero. But in practice, due to the limited numerical range and the non-linearities such as ReLU in LLMs, $diag(\mathbf{x})$ can become

---

**Algorithm 2** SLIM Saliency-based Low-rank Adapter Computation

---

1: **Input:** Original Weight: $\mathcal{W}$, Compressed Weight: $\mathcal{W}^C$ Calibration Input: $\mathcal{X}$
2: **Output:** $\mathcal{L}, \mathcal{R}$: Saliency-based Low-rank Adapters
3: $E_C = E_Q + E_S = \mathcal{W}^C - \mathcal{W}$     *// Compute compression Error*
4: $\tilde{\mathbf{x}} = mean(\mathcal{X})$     *// Average over all the samples*
5: $\mathbf{x} = \tilde{\mathbf{x}} + min(|\tilde{\mathbf{x}}|)$     *// Shift all the values to avoid zeros in $\mathbf{x}$*
6: $\mathcal{S}_C = diag(\mathbf{x})E_C$     *// Compute compression error saliency*
7: $\tilde{\mathcal{L}}, \tilde{\mathcal{R}} = SVD(\mathcal{S}_C)$     *// Low-rank approximation of compression error saliency*
8: $\mathcal{L} = diag(1/\mathbf{x})\tilde{\mathcal{L}}, \mathcal{R} = \tilde{\mathcal{R}}$     *// Converting saliency factors to weight factors*

---

singular. To avoid such cases. This behavior will also lead to having the rows of the saliency matrix to be set to zero, not distinguishing between the less and more important weights in a row. To overcome these challenges, we have added the minimum absolute value available in $\mathbf{x}$ to all its elements. Algorithm 2 summarizes all the details of computing the saliency-based low-rank adapters in SLIM . Please note that we have used the additive property of the saliency function to compute the saliency of the errors separately, and optimize the low-rank adapters to minimize the error saliency. The Wanda saliency holds the additive property, making it suitable for our work, while other methods that use the second-order information of the loss function cannot utilize saliency-based low-rank adapters.

### 3.3 POST-COMPRESSION FINE-TUNING

Fine-tuning models after applying one-shot compression presents significant challenges, primarily due to the limitations imposed by the integer representation of parameters in quantized weights. Quantized weights have limited precision and restricted value ranges, making gradient-based updates difficult and potentially leading to loss of information during fine-tuning. Moreover, the high parameter count of large language models renders traditional fine-tuning extremely computationally expensive and time-consuming, necessitating more parameter-efficient methods.

SLIM addresses these issues by introducing fine-tunable parameters in the form of low-rank adapters. In its optional fine-tuning phase, SLIM freezes the sparse and quantized weights, allowing only the tuning of these low-rank adapters. This parameter-efficient fine-tuning approach enables rapid improvement in the compressed model's accuracy using a short fine-tuning phase over just thousands of tokens. By focusing the fine-tuning process on a small subset of parameters (the adapters), SLIM significantly reduces the computational requirements while still allowing the model to adapt to new data or tasks. This approach strikes a balance between maintaining the benefits of compression and enabling post-compression adaptation.

### 3.4 TILED LOW-RANK ADAPTER QUANTIZATION

Pruning and quantizing the weights reduces the computation and memory footprint of the models significantly ($\sim 8\times$ reduction in memory size), but adding low-rank adapters in full precision will result in an extra overhead. To reduce the adapter overheads, we compress the adapters using quantization. The distribution of the elements in the factors have long tails, making even advanced methods that don't use group quantization such as SLIM-Quant impractical. On the other hand, available group quantization methods use 1-dimensional tiles for quantization, which does not match the layout used for tensor cores, hence making them not hardware-friendly.

To address these issues, we propose a tiled quantization scheme for the low-rank adapters, in which 256 elements of the adapter are quantized using the same quantization parameter, creating $16 \times 16$ tiles. The choice of $16 \times 16$ blocks is made based on the input size of tensor cores in NVIDIA A100 and H100 GPUs, making it the tiling strategy more hardware friendly and allowing the warps in the GPU to dequantize the data in for different tensor cores in parallel using only one quantization parameter per tensor core. The quantization of each tile is done using the AbsMax algorithm.

# 4 EXPERIMENTAL RESULTS

**Models, Datasets, and Evaluation.** We evaluate SLIM on the OPT (Zhang et al., 2022) and LLaMA-2 (Touvron et al., 2023) model families, both of which serve as standard baselines in model compression studies (Frantar et al., 2022; Frantar & Alistarh, 2023; Sun et al., 2023). Model accuracy is assessed on a range of zero-shot downstream tasks, including MMLU (Hendrycks et al., 2020), Piqa (Bisk et al., 2020), Arc-Easy, Arc-Challenge (Clark et al., 2018), WinoGrande (Sakaguchi et al., 2021), and OpenBookQA (Mihaylov et al., 2018). For zero-shot evaluations, we utilize the Language Model Evaluation Harness (Gao et al., 2024) framework. In line with prior work (Sun et al., 2023; Frantar & Alistarh, 2023; Frantar et al., 2022), we also report the perplexity of the models on a language modeling task on the WikiText2 (Merity et al., 2016) dataset, provided in Appendix A.

**Baselines.** We compare SLIM against state-of-the-art one-shot pruning methods, including Wanda (Sun et al., 2023), SparseGPT (Frantar & Alistarh, 2023), and Magnitude Pruning (Han et al., 2015), as well as one-shot quantization techniques like OPTQ (Frantar et al., 2022) and AbsMax. Since AWQ (Lin et al., 2024) relies on floating point activations, we have not included it in our experiments, in which both weights and activations are quantized.

Similar to other leading one-shot pruning and quantization methods (Wanda, SparseGPT, OPTQ), SLIM leverages calibration data to extract statistics and assess weight saliency. As Wanda, SparseGPT, and OPTQ operate under identical conditions, we adopt their approach, using 128 sequences sampled from the C4 (Raffel et al., 2019) dataset. Additionally, for all fine-tuning experiments, we utilize 300,000 tokens from the C4 dataset. SLIM uses Wanda for pruning the models, and the Wanda saliency is further used for optimizing the low-rank adapters.

**Sparse and Quantized.** We evaluate model accuracy across structured and unstructured sparsity benchmarks for various pruning and quantization methods. Specifically, we pair OPTQ and SparseGPT, which follow similar error recovery strategies, and combine Wanda with AbsMax and SLIM-Quant , while using Magnitude Pruning with AbsMax. To demonstrate the effectiveness of SLIM saliency-based low-rank adapters, we introduce low-rank adapters to Wanda, minimizing the magnitude of the error (rather than the saliency) using SVD. This approach, referred to as the "SVD low-rank adapter or Wanda-SVD" in the tables, is similar to LQLoRA (Guo et al., 2023) with the difference that LQLoRA does not prune the weights, nor uses SLIM-Quant for quantization, leading to significantly higher error rates. We do not apply one-shot low-rank adapters to SparseGPT or OPTQ, as their weight update rules conflict with minimizing weight error. Additionally, for a more thorough experiment setting, we have implemented group quantization for AbsMax, and have tested it with different settings.

Table 1 presents the zero-shot task accuracy results for sparse and quantized versions of the OPT and LLaMA-2 models. Among the tested methods, Magnitude Pruning combined with AbsMax quantization yields the lowest accuracy, with AbsMax applied to Wanda showing similarly poor performance. While applying SLIM-Quant to Wanda alleviates some of the accuracy loss, it still falls short compared to SparseGPT and OPTQ. In contrast, SLIM achieves the highest accuracy across all methods, with further improvements gained through a brief fine-tuning step (SLIM + FT). Additionally, SLIM$^Q$ quantizes the low-rank adapters to 4 bits, reducing adapter overhead by $4\times$ while maintaining competitive accuracy. Notably, while Group AbsMax improves the accuracy of models with low-rank adapters, it still underperforms compared to SLIM-Quant , which employs a single quantization parameter per tensor for greater efficiency.

A similar trend is observed for unstructured sparsity, although the performance gap between dense and sparse models is smaller across all methods. SLIM achieves a significant improvement in average accuracy, boosting results by up to 5.4% (LLaMA-2-7B) for 2:4 sparsity compared to the state-of-the-art SparseGPT and Group OPTQ. This gap further widens to 5.8% (LLaMA-2-13B) when incorporating an additional parameter-efficient fine-tuning step.

**Sparse Only.** To isolate the effects of sparsity on model accuracy, we conduct a series of experiments with quantization disabled. Our sparsity benchmarks include Magnitude Pruning, SparseGPT, and Wanda, along with low-rank approximations using Wanda-SVD and SLIM . We evaluate both 50% unstructured sparsity and 2:4 structured sparsity patterns in our experiments.

Table 1: Average zero-shot accuracy of LLaMA-2 and OPT models with pruning, 4-bit symmetric weight quantization, and 8-bit symmetric input group quantization. Wanda-SVD uses SVD directly on the compression error matrix, and Wanda-SVD + FT and SLIM + FT uses fine-tuning on low-rank adapters for 300,000 tokens. SLIM$^Q$ quantizes the low-rank adapters after the compression (and possibly fine-tuning) process.

| Pruning Method | Weight Quantization | OPT | | | | | | LLaMA-2 | |
|---|---|---|---|---|---|---|---|---|---|
| | | 125M | 350M | 1.3B | 2.7B | 6.7B | 13B | 7B | 13B |
| Dense | - | 35.9 | 37.1 | 43.4 | 45.5 | 48.3 | 48.7 | 56.6 | 60.8 |
| **50% 2:4** | | | | | | | | | |
| Magnitude | AbsMax | 32.0 | 31.8 | 34.2 | 32.5 | 35.3 | 30.8 | 31.2 | 32.1 |
| SparseGPT | Group-OPTQ | 33.7 | 32.6 | 37.3 | 40.2 | 44.4 | 45.5 | 45.4 | 50.8 |
| SparseGPT | OPTQ | 31.4 | 32.9 | 31.0 | 33.9 | 39.9 | 40.0 | 31.8 | 31.6 |
| Wanda | Group AbsMax | 33.0 | 31.6 | 36.3 | 35.1 | 36.6 | 43.4 | 43.1 | 48.3 |
| Wanda | AbsMax | 31.5 | 31.3 | 31.6 | 30.7 | 30.5 | 31.2 | 32.0 | 31.3 |
| Wanda | SLIM-Quant | 31.8 | 32.1 | 34.7 | 34.3 | 38.4 | 32.8 | 30.8 | 30.7 |
| Wanda-SVD | Group AbsMax | 33.9 | 34.0 | 38.9 | 39.9 | 44.2 | 45.5 | 50.5 | 54.5 |
| Wanda-SVD | SLIM-Quant | 34.2 | 33.3 | 38.7 | 41.2 | 44.3 | 45.2 | 48.3 | 51.4 |
| Wanda-SVD + FT | SLIM-Quant | 34.0 | 34.3 | 39.6 | 42.6 | **46.1** | 47.2 | 50.8 | 55.4 |
| SLIM | Group AbsMax | 33.9 | 33.7 | 39.9 | 42.8 | 45.8 | 46.0 | 50.2 | 54.3 |
| SLIM | SLIM-Quant | 34.3 | 33.5 | 40.0 | 42.8 | **46.1** | 46.1 | **50.8** | 54.8 |
| SLIM$^Q$ | SLIM-Quant | 34.2 | 33.8 | 39.8 | 41.8 | 46.0 | 45.9 | 50.6 | 53.0 |
| SLIM + FT | SLIM-Quant | **34.9** | **34.5** | **41.3** | **43.5** | **46.1** | **47.3** | 50.5 | **56.6** |
| SLIM$^Q$ + FT | SLIM-Quant | **34.9** | 34.3 | 40.0 | 42.3 | 46.0 | 46.5 | 50.6 | 54.1 |
| **50% Unstructured** | | | | | | | | | |
| Magnitude | AbsMax | 31.1 | 32.9 | 33.1 | 36.2 | 36.3 | 31.2 | 32.6 | 31.5 |
| SparseGPT | Group-OPTQ | 35.1 | 35.1 | 38.9 | 43.2 | 47.1 | 47.3 | 50.1 | 55.4 |
| SparseGPT | OPTQ | 31.4 | 34.5 | 31.2 | 37.1 | 43.2 | 44.1 | 31.7 | 32.0 |
| Wanda | Group AbsMax | 34.2 | 33.3 | 39.1 | 40.7 | 44.9 | 46.2 | 51.7 | 55.8 |
| Wanda | AbsMax | 31.5 | 32.9 | 31.0 | 32.9 | 30.5 | 31.1 | 32.7 | 31.1 |
| Wanda | SLIM-Quant | 32.8 | 33.9 | 36.0 | 36.2 | 42.7 | 32.8 | 30.4 | 30.5 |
| Wanda-SVD | Group AbsMax | 34.6 | 34.4 | 40.5 | 42.9 | 46.3 | 47.2 | 53.9 | 55.4 |
| Wanda-SVD | SLIM-Quant | 34.6 | 34.4 | 40.3 | 43.3 | 46.7 | 45.2 | 51.2 | 55.4 |
| Wanda-SVD + FT | SLIM-Quant | 35.3 | 34.8 | 41.8 | 43.8 | 47.0 | 47.9 | 53.0 | 57.3 |
| SLIM | Group AbsMax | 35.0 | 35.0 | 41.5 | 43.6 | 47.2 | 47.9 | **54.0** | **57.6** |
| SLIM | SLIM-Quant | **35.7** | 35.4 | 42.0 | 43.4 | 47.5 | 48.0 | **54.0** | **57.6** |
| SLIM$^Q$ | SLIM-Quant | 34.8 | 35.0 | 41.4 | 34.3 | 47.1 | 47.4 | 53.8 | 57.1 |
| SLIM + FT | SLIM-Quant | **35.7** | **35.8** | **42.3** | **44.3** | 47.3 | **48.4** | 53.2 | 57.0 |
| SLIM$^Q$ + FT | SLIM-Quant | 35.3 | 35.6 | 41.9 | 43.8 | **47.6** | 48.1 | 53.7 | **57.6** |

Table 2 presents the accuracy results for the sparse models. As anticipated, Magnitude Pruning yields the lowest accuracy. Wanda and SparseGPT achieve comparable results, though in the case of semi-structured sparsity, their performance gap with the dense model is more pronounced. Introducing low-rank adapters improves model accuracy, with SLIM being particularly effective due to its saliency-based approximation. Finally, a brief fine-tuning phase further enhances the accuracy of the low-rank approximations.

**Quantized Only.** To assess the effects of SLIM-Quant and the low-rank compensation in SLIM , we disable sparsity in our experiments and evaluate various quantization schemes. Specifically, we test AbsMax, OPTQ, and SLIM-Quant as the quantization methods. To improve model accuracy,

Table 2: Average zero-shot accuracy of LLaMA-2 and OPT models with pruning. The quantization is disabled in this experiment.

| Pruning Method | OPT | | | | | | LLaMA-2 | |
|---|---|---|---|---|---|---|---|---|
| | 125M | 350M | 1.3B | 2.7B | 6.7B | 13B | 7B | 13B |
| Dense | 35.9 | 37.1 | 43.4 | 45.5 | 48.3 | 48.7 | 56.6 | 60.8 |
| **50% 2:4** | | | | | | | | |
| Magnitude | 32.6 | 31.8 | 35.4 | 33.9 | 36.4 | 30.7 | 31.2 | 32.0 |
| SparseGPT | 33.8 | 33.2 | 37.7 | 41.3 | 45.2 | 45.6 | 47.3 | 52.3 |
| Wanda | 34.0 | 32.5 | 38.3 | 40.5 | 43.2 | 44.1 | 46.1 | 49.7 |
| Wanda-SVD | 34.1 | 34.1 | 40.4 | 42.8 | 46.0 | 45.9 | 51.6 | 55.8 |
| Wanda-SVD + FT | 34.8 | 34.5 | 41.3 | 43.4 | **46.5** | 47.2 | **52.4** | **56.9** |
| SLɪM | 34.5 | 32.9 | 40.7 | 43.1 | 46.4 | 46.3 | 51.4 | 56.1 |
| SLɪM + FT | **35.1** | **34.9** | **41.5** | **43.8** | **46.5** | **47.3** | 51.6 | 56.4 |
| **50% Unstructured** | | | | | | | | |
| Magnitude | 33.3 | 33.7 | 34.0 | 40.6 | 35.8 | 30.9 | 32.6 | 31.9 |
| SparseGPT | 35.5 | 35.1 | 39.6 | 43.5 | 47.4 | 47.8 | 53.3 | 57.3 |
| Wanda | 35.0 | 34.5 | 41.1 | 42.9 | 46.5 | 46.8 | 52.7 | 57.2 |
| Wanda-SVD | 35.3 | 35.2 | 41.9 | 44.1 | 47.5 | 47.8 | 54.9 | 58.5 |
| Wanda-SVD + FT | 35.74 | 35.7 | 42.7 | 44.6 | **47.8** | **48.4** | 54.9 | 58.7 |
| SLɪM | 35.2 | 35.1 | 42.0 | 44.1 | 47.7 | 48.2 | **55.0** | 58.8 |
| SLɪM + FT | **35.9** | **35.7** | **42.5** | **44.7** | 47.7 | **48.4** | **55.0** | **58.8** |

we add low-rank adapters to SLɪM-Quant , minimizing either the error saliency (SLɪM ) or the reconstruction error norm (SVD). Low-rank adapters cannot be applied to OPTQ due to its weight update rules, which conflict with minimizing the weight reconstruction error.

Table 3 summarizes the results of our quantization experiments. As expected, AbsMax produces the highest error among the quantization methods. While SLɪM-Quant addresses some of the limitations of AbsMax, it still struggles to fully recover from accuracy loss. OPTQ achieves higher accuracy than both AbsMax and SLɪM-Quant , but its inability to incorporate low-rank adapters limits its effectiveness at lower bitwidths. Both the SVD and SLɪM low-rank adapters enhance the accuracy of SLɪM-Quant , with SLɪM outperforming SVD due to its saliency-based approximation.

Table 3: Average zero-shot accuracy of LLaMA-2 and OPT models with quantization. The sparsity is disabled in this experiment.

| Quantization Method | Low-rank Adapter | OPT | | | | | | LLaMA-2 | |
|---|---|---|---|---|---|---|---|---|---|
| | | 125M | 350M | 1.3B | 2.7B | 6.7B | 13B | 7B | 13B |
| Dense | - | 35.9 | 37.1 | 43.4 | 45.5 | 48.3 | 48.7 | 56.6 | 60.8 |
| AbsMax | - | 30.7 | 34.0 | 31.2 | 31.6 | 30.3 | 31.4 | 32.4 | 31.9 |
| Group-OPTQ | - | 35.5 | 36.2 | 42.5 | 44.5 | 47.7 | 48.2 | 53.3 | 59.6 |
| OPTQ | - | 31.4 | 36.0 | 31.5 | 37.3 | 43.3 | 45.1 | 31.2 | 31.5 |
| SLɪM-Quant | - | 32.0 | 36.5 | 36.2 | 40.0 | 30.3 | 37.8 | 31.0 | 30.5 |
| SLɪM-Quant | SVD | 35.5 | 35.9 | 42.4 | 44.9 | 47.8 | 48.2 | 55.8 | 60.5 |
| SLɪM-Quant | SVD + FT | **35.9** | 36.4 | **43.2** | 45.5 | 48.2 | 48.7 | **56.0** | 60.4 |
| SLɪM-Quant | SLɪM | 35.6 | **36.5** | 42.7 | 45.4 | **48.3** | 48.4 | **56.0** | 60.2 |
| SLɪM-Quant | SLɪM + FT | 35.7 | **36.5** | **43.2** | 45.6 | **48.3** | 48.9 | 55.8 | **60.4** |

**Fine-tuning Costs.** Fine-tuning compressed models can help recover lost accuracy. However, fine-tuning quantized weights presents challenges due to the discrete nature of the weights. The most commonly used approach for addressing these challenges, and one that has shown promising results, is the straight-through estimator (STE) (Bengio et al., 2013). In this method, during the backward pass and optimization step, the weights are treated as continuous, allowing for effective fine-tuning despite the quantization.

In addition to the challenges posed by discrete values during fine-tuning, the high parameter count of the models leads to time-consuming computations and substantial memory costs. In our experiments, we measured the time required to fine-tune the models under various conditions. For models with low-rank adapters, the quantized weights are frozen, allowing only the low-rank adapters to be fine-tuned. This approach results in a more parameter-efficient fine-tuning strategy, reducing both memory and computational costs. When no low-rank adapter is employed, the straight-through estimator (STE) is used for fine-tuning the quantized weights. Table 4 summarizes the fine-tuning results for 300,000 tokens from the C4 dataset, with a batch size of 64 and a sequence length of 1024 on a single H100 GPU. The fine-tuning costs for models without low-rank adapters range from 12 hours for 125M parameter models to over 36 days for 13B parameter models. Due to these high costs, we faced challenges completing the fine-tuning with our limited resources. In contrast, utilizing low-rank adapters and freezing the sparse quantized weights enables a much more parameter-efficient fine-tuning method, making it feasible for us to report the accuracy results for these cases in Table 1.

Table 4: The required time for fine-tuning the models with a single H100 GPU on 300,000 tokens from the C4 dataset with a batch size of 64 and a sequence length of 1024.

| Pruning Method | Weight Quantization | OPT | | | | | | LLaMA-2 | |
|---|---|---|---|---|---|---|---|---|---|
| | | 125M | 350M | 1.3B | 2.7B | 6.7B | 13B | 7B | 13B |
| Magnitude SparseGPT Wanda | AbsMax OPTQ AbsMax | 12h | 43h | 164h | 361h | 866h | 867h | 842h | 844h |
| Wanda-SVD SLiM + FT | SLiM-Quant SLiM-Quant | 1.5h | 3h | 6h | 8h | 16h | 18h | 14h | 14h |

**Additional Experiments.** In the appendix, we provide additional experiments for a comprehensive evaluation. **Compression Costs** (Appendix F) examines the time required to compress models of varying sizes using different methods. **Inference Speedup** (Appendix G) evaluates the performance gains during inference, highlighting the efficiency improvements achieved by our approach. **Rank Analysis** (Appendix H) investigates the computational and memory costs of different ranks in low-rank adapter methods, along with their impact on model accuracy. Finally, **Effects of Calibration Sample Count** (Appendix I) analyzes how varying the number of calibration samples affects accuracy in methods that require calibration. Finally, we have included additional experiments with other pruning quantization techniques in Appendix B.

## 5 Conclusion, Limitations, and Future Work

In this paper, we introduced SLiM, a one-shot quantized sparse plus low-rank approximation method for large language models, designed to balance memory efficiency and accuracy. By leveraging symmetric quantization, sparsity, and saliency-based low-rank adapters, SLiM achieves significant reductions in both memory and computational costs while maintaining competitive performance. Our method demonstrates improved accuracy, particularly for models with structured sparsity patterns like 2:4 sparsity, compared to state-of-the-art approaches.

SLiM relies on current available libraries all of which lack support for advanced quantization schemes like 2:4 mixed 8-bit and 4-bit group quantization. Additionally, low-rank adapters, while effective, introduce overheads. Future work will focus on developing efficient kernels for 2:4 group quantization and compressing low-rank adapters to further optimize memory and speed.

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

# Appendix

## Table of Contents

## A  LANGUAGE MODELING EXPERIMENTS

We have tested all the benchmarks discussed in Section 4 on a language modeling task on the Wiki-Text2 (Merity et al., 2016) dataset. Table 5 summarizes the results for different pruning and quantization approaches when using 4-bit weight and 8-bit group input quantization. Similar to Section 4, SLιM outperforms all the previous methods, including SparseGPT with Group OPTQ. Using saliency based methods for low-rank adapters is also improving the perplexity of the models in comparison to Wanda-SVD. Additionally, a short fine-tuning step can improve the perplexity of the models significantly.

## B  ADDITIONAL QUANTIZATION METHODS

We have tested additional quantization and pruning methods such as Joint Sparsity and Quantization (JSQ) (Guo et al., 2024), OmniQuant (Shao et al., 2023), AffineQuant (Ma et al., 2024), and AWQ (Lin et al., 2024) and compared it against SLιM on WikiText2 (Merity et al., 2016) language modeling task. Table ?? summarizes the perplexity results of all the methods. In all these experiments (except for JSQ), the weight is pruned with Wanda and quantized to 4 bits. The inputs are quantized to 8 bit as well. The optimal parameters for each model are used if the papers reported them, otherwise, the default quantization parameters are used.

As shown in table 6, all the quantization and pruning methods resulted in higher errors compared to SLιM (and even SparseGPT + OPTQ), as they were not co-designed with the pruning method. This highlights the importance of co-designing pruning, quantization, and low-rank adapters to achieve optimal performance.

Furthermore, methods like JSQ demonstrate strong performance in only one class of models but struggle to generalize to others, such as the OPT models. Similarly, techniques like OmniQuant, AffineQuant, and AWQ fail to effectively quantize the OPT-350M model, often resulting in NaN (Not a Number) or extremely high perplexity values. Moreover, AffineQuant faces challenges quantizing the sparse version of LLaMA-2-7B model due to the large recovery loss in the last two layers of the models. All experiments were conducted on a single A100-40GB GPU; notably, AWQ encountered out-of-memory (OOM) issues when applied to the LLaMA-2-13B model, as indicated in the table.

Table 5: Perplexity of LLaMA-2 and OPT models with pruning, 4-bit symmetric weight quantization, and 8-bit symmetric input group quantization on WikiText2 dataset language modeling task. Wanda-SVD uses SVD directly on the compression error matrix, and Wanda-SVD + FT and SLiM + FT uses fine-tuning on low-rank adapters for 300,000 tokens.

| Pruning Method | Weight Quantization | OPT | | | | | | LLaMA-2 | |
|---|---|---|---|---|---|---|---|---|---|
| | | 125M | 350M | 1.3B | 2.7B | 6.7B | 13B | 7B | 13B |
| Dense | - | 27.7 | 22.0 | 14.6 | 12.5 | 10.9 | 10.1 | 5.1 | 4.6 |
| **50% 2:4** | | | | | | | | | |
| Magnitude | AbsMax | 6.2e3 | 9.1e2 | 2.1e3 | 1.5e3 | 436.3 | 4.1e4 | 9.1e4 | 1.1e5 |
| SparseGPT | Group-OPTQ | 78.6 | 62.3 | 27.1 | 18.7 | 15.4 | 12.9 | 15.2 | 8.3 |
| SparseGPT | OPTQ | 6.7e3 | 77.2 | 762.3 | 52.1 | 21.1 | 15.36 | 7.4 | 1.2e5 |
| Wanda | AbsMax | 6.2e3 | 412.6 | 1.0e4 | 1.0e4 | 8.8e3 | 8.2e3 | 8.1e4 | 6.6e4 |
| Wanda | SLiM-Quant | 308.4 | 145.7 | 1.3e3 | 441.5 | 65.2 | 2.3e3 | 5.2e5 | 7.6e4 |
| Wanda-SVD | SLiM-Quant | 83.6 | 60.7 | 27.4 | 21.3 | 14.7 | 13.2 | 8.2 | 7.1 |
| Wanda-SVD + FT | SLiM-Quant | 54.5 | 42.1 | 21.1 | 16.3 | 13.4 | 12.6 | 7.0 | 6.0 |
| SLiM | SLiM-Quant | 58.1 | 51.7 | 20.0 | 15.8 | 12.8 | 12.1 | 7.6 | 6.6 |
| SLiM + FT | SLiM-Quant | **41.5** | **36.3** | **17.6** | **14.7** | **12.4** | **12.1** | **6.4** | **5.8** |
| **50% Unstructured** | | | | | | | | | |
| Magnitude | AbsMax | 3.1e3 | 127.3 | 7.1e3 | 1.1e3 | 772.3 | 2.3e4 | 9.9e4 | 2.0e5 |
| SparseGPT | Group-OPTQ | 42.6 | 34.7 | 20.3 | 14.4 | 12.2 | 11.3 | 8.4 | 5.6 |
| SparseGPT | OPTQ | 4.8e3 | 39.4 | 463.0 | 29.7 | 15.1 | 12.7 | 1.2e4 | 8.6e4 |
| Wanda | AbsMax | 5.9e3 | 84.2 | 1.3e4 | 1.1e3 | 7.1e3 | 5.6e3 | 9.3e4 | 1.6e5 |
| Wanda | SLiM-Quant | 136.9 | 57.9 | 460.9 | 176.0 | 56.5 | 623.3 | 2.9e5 | 7.1e4 |
| Wanda-SVD | SLiM-Quant | 47.0 | 34.5 | 19.6 | 15.5 | 12.3 | 11.5 | 6.6 | 5.6 |
| Wanda-SVD + FT | SLiM-Quant | 39.2 | 29.7 | 17.8 | 14.3 | 12.3 | 11.9 | 6.1 | 5.3 |
| SLiM | SLiM-Quant | 39.7 | 32.0 | 16.7 | 13.7 | **11.5** | **10.8** | 6.2 | 5.4 |
| SLiM + FT | SLiM-Quant | **34.0** | **28.3** | **15.9** | **13.4** | **11.5** | 11.5 | **5.8** | **5.2** |

It is noteworthy that none of the quantization methods we tested can be combined with saliency-based low-rank adapter methods such as SLiM or Wanda-SVD. This is because these quantization methods alter the values of the weights during the quantization process, which disrupts the saliency-based optimization of low-rank adapters. This further emphasizes the importance of co-designing pruning, quantization, and low-rank adapters to ensure compatibility and maintain optimal performance.

## C  MEMORY REDUCTION ANALYSIS

SLiM prunes and quantizes the models and adds additional low-rank adapters to them. Additionally, it supports quantization methods for the low-rank adapters to reduce their overheads. In the following, we propose an analysis on the reduced memory when using SLiM and other pruning and quantization methods.

Assuming the hidden dimension of a model is $d$ and the low-rank adapter ratio used in the model is of rank $r < 1$. Furthermore, by denoting the number of transformer blocks with $n$ and the vocabulary size of the model by $V$ and by denoting the ratio of the up-projection and down-projection layers in the model by $a$, we can get the memory reduction as the ratio of $\frac{\text{Compressed Model Size}}{\text{Dense Model Size}}$ from equation 13.

$$\text{Memory Reduction} = \frac{n(4d^2 + 2d^2a) + dV}{n(4d^2/2 + 4 \times 2d^2r + 2d^2a/2 + 2d(dr + dra)) + dV} \tag{13}$$

Table 6: Perplexity of LLaMA-2 and OPT models with 2:4 and unstructured sparsity pruning, evaluated with various weight quantization schemes on the WikiText2 language modeling task. SLiM + FT employs fine-tuning on low-rank adapters for 300,000 tokens.

| Pruning Method | Quantization | OPT 125M | 350M | 1.3B | 2.7B | 6.7B | 13B | LLaMA-2 7B | 13B |
|---|---|---|---|---|---|---|---|---|---|
| **Dense** | - | 27.66 | 22.00 | 14.62 | 12.47 | 10.86 | 10.13 | 5.12 | 4.57 |
| **2:4 Sparsity** | | | | | | | | | |
| JSQ | JSQ | 3.21e3 | 1.97e4 | 71.97 | 2.87e3 | 23.76 | 596.60 | 11.35 | 8.25 |
| Wanda | OmniQuant | 97.12 | NaN | 34.86 | 27.03 | 18.91 | 18.06 | 12.78 | 6.00 |
| Wanda | AWQ | 83.02 | 7.30e5 | 27.88 | 21.60 | 17.93 | 16.72 | 12.79 | OOM |
| Wanda | AffineQuant | 86.58 | NaN | 28.30 | 21.99 | 16.62 | 16.83 | 6.59e3 | 6.42e3 |
| SLiM | SLiM-Quant | 58.10 | 51.70 | 20.00 | 15.80 | 12.80 | 12.10 | 7.60 | 6.60 |
| SLiM + FT | SLiM-Quant | **41.50** | **36.30** | **17.60** | **14.70** | **12.40** | **12.10** | **6.40** | **5.80** |
| **Unstructured Sparsity** | | | | | | | | | |
| JSQ | JSQ | 3.07e3 | 2.99e4 | 24.91 | 250.99 | 15.44 | 2.30e5 | 6.66 | 5.69 |
| Wanda | OmniQuant | 44.90 | NaN | 20.59 | 15.70 | 13.12 | 13.89 | 7.39 | 6.36 |
| Wanda | AWQ | 39.23 | 3.41e5 | 17.97 | 14.16 | 11.89 | 12.34 | 7.28 | OOM |
| Wanda | AffineQuant | 40.91 | NaN | 18.53 | 14.55 | 12.18 | 12.50 | 1.08e4 | 7.69e3 |
| SLiM | SLiM-Quant | 39.70 | 32.00 | 16.70 | 13.70 | 11.50 | 10.80 | 6.20 | 5.40 |
| SLiM + FT | SLiM-Quant | **34.00** | **28.30** | **15.90** | **13.40** | **11.50** | **11.50** | **5.80** | **5.20** |

Table 7 summarizes the memory reduction of different pruning and quantization methods. Please note that when using low-rank adapters (in Wanda-SVD and SLiM ), we assume a rank of $r = 0.1$.

Table 7: Memory reduction ($\times$) of different compression methods across various OPT and LLaMA models. In Quantized SLiM , the low-rank adapters are also quantized.($\downarrow$ is better.)

| Compression Method | OPT 125M | 350M | 1.3B | 2.7B | 6.7B | 13B | LLaMA-2 7B | 13B |
|---|---|---|---|---|---|---|---|---|
| SparseGPT + OPTQ | 0.40 | 0.30 | 0.25 | 0.17 | 0.15 | 0.14 | 0.15 | 0.14 |
| Wanda + AbsMax | 0.40 | 0.30 | 0.25 | 0.17 | 0.15 | 0.14 | 0.15 | 0.14 |
| Wanda-SVD + AbsMax | 0.50 | 0.42 | 0.38 | 0.31 | 0.30 | 0.29 | 0.31 | 0.30 |
| SLiM + SLiM -Quant | 0.50 | 0.42 | 0.38 | 0.31 | 0.30 | 0.29 | 0.31 | 0.30 |
| Quantized SLiM + SLiM -Quant | 0.42 | 0.33 | 0.28 | 0.20 | 0.19 | 0.18 | 0.19 | 0.18 |

## D   COMPUTATION REDUCTION ANALYSIS

SLiM and other compression method reduce the number of floating point operations (FLOPs) at the inference of models. Additionally, the low-rank adapters used in SLiM and Wanda SVD can add additional computational overheads to the inference of the models. Following JSQ (Guo et al., 2024), in this section, we provide an analysis on the FLOP reduction in the inference of different methods. It is noteworthy that even though quantization can reduce the memory overhead of models, since all the computations are done in floating point format, it does not lead to a reduction in the computation of the inference.

Assuming the hidden dimension of a model is $d$ and the low-rank adapter ratio used in the model is of rank $r < 1$. Furthermore, by denoting the number of transformer blocks with $n$ and the vocabulary size of the model by $V$ and by denoting the ratio of the up-projection and down-projection layers in the model by $a$, we can get the memory reduction as the ratio of $\frac{\text{Dense Inference FLOP Count}}{\text{Compressed Inference FLOP Count}}$ from equation 14, where $b$ is the batch size, and is canceled in the numerator and the denominator of the equation.

$$\text{FLOP Reduction} = \frac{n(4bd^2 + 2bd^2a) + bdV}{n(4bd^2/2 + 4 \times 2bd^2r + 2bd^2a/2 + 2b(d^2r + d^2ra)) + bdV} \tag{14}$$

Table 8 summarizes the FLOP reduction of different compression methods. As it can be seen, the overhead of adding the low-rank adapters ($r = 0.1$) in SLIM and Wanda-SVD is not significant.

Table 8: Compute (FLOP) reduction ratios ($\times$) of different compression methods across various OPT and LLaMA models. In Quantized SLIM , the low-rank adapters are also quantized. ($\uparrow$ is better.)

| Compression | OPT | | | | | | LLaMA-2 | |
| Method | 125M | 350M | 1.3B | 2.7B | 6.7B | 13B | 7B | 13B |
| --- | --- | --- | --- | --- | --- | --- | --- | --- |
| SparseGPT + OPTQ | 1.52 | 1.66 | 1.75 | 1.91 | 1.94 | 1.96 | 1.95 | 1.97 |
| Wanda + AbsMax | 1.52 | 1.66 | 1.75 | 1.91 | 1.94 | 1.96 | 1.95 | 1.97 |
| Wanda-SVD + AbsMax | 1.32 | 1.39 | 1.43 | 1.50 | 1.51 | 1.52 | 1.49 | 1.49 |
| SLIM + SLIM -Quant | 1.32 | 1.39 | 1.43 | 1.50 | 1.51 | 1.52 | 1.49 | 1.49 |
| Quantized SLIM + SLIM -Quant | 1.32 | 1.39 | 1.43 | 1.50 | 1.51 | 1.52 | 1.49 | 1.49 |

## E    FINE-TUNING HYPERPARAMETERS

For fine-tuning the models, we utilized the Hugging Face Trainer (Wolf, 2019). The ADAMW (Loshchilov, 2017) optimizer was employed during the fine-tuning process, accompanied by linear learning rate scheduling. The optimization and learning rate scheduling parameters were set to their default values in the Hugging Face Trainer. To prevent numerical overflow and divergence, we used BFloat-16 data types (Wang & Kanwar) available on NVIDIA H100 GPUs during fine-tuning. The training was conducted with a local batch size of 1 and a gradient accumulation factor of 64 to reduce memory overhead. Weight updates for the sparse and/or quantized weights, as well as the corresponding biases, were disabled. Due to our limited resources, we did not tune any of the hyperparameters aimed at improving fine-tuning speed or accuracy; tuning these parameters is planned for future work.

## F    COMPRESSION COSTS

An important factor in model compression is the computational cost of the chosen method. In terms of memory usage, all approaches can be adapted to store only a single layer of the model in the GPU's global memory at a time, allowing them to be compressed on a single GPU. However, the computational costs vary depending on the complexity of the method. Techniques like Wanda, which rely solely on matrix multiplication, are significantly faster than more complex methods like SparseGPT, which computes the inverse Hessian matrix for each layer. Adding low-rank adapters in Wanda-SVD and SLIM requires performing singular value decomposition (SVD) on different matrices, resulting in a computational complexity similar to that of SparseGPT. Table 9 summarizes the time required to compress various models using the methods discussed in this paper. Generally, methods incorporating low-rank adapters (SLIM and Wanda-SVD) have higher complexity. However, SparseGPT's compression time is comparable to methods with low-rank adapters, despite only performing pruning and quantization. Notably, the saliency-based approach in SLIM does not add significant overhead compared to Wanda-SVD.

## G    INFERENCE SPEEDUP

Many libraries, such as CUTLASS (NVIDIA Corporation, b) and cuSPARSELt (NVIDIA Corporation, a), have implemented sparse matrix-matrix multiplication (SpMM) kernels for 2:4 sparsity. However, to the best of our knowledge, there is no open-source code base that supports group quantization for 2:4 SpMM. Implementing such a kernel is beyond the scope of this paper, and we propose its development as part of future work. Similar to Wanda (Sun et al., 2023) and SparseGPT (Frantar & Alistarh, 2023), we focus our results on layer-wise speedups achieved using the existing code bases. Table 10 summarizes the time taken for dense, sparse, and low-rank multiplication in a linear layer, and reports the speedup achieved by SLIM across different layers in the LLaMA-2-13B model.

Table 9: The required compresion time for different models and compression methods using a single H100 GPU.

| Pruning Method | Weight Quantization | OPT | | | | | | LLaMA-2 | |
|---|---|---|---|---|---|---|---|---|---|
| | | 125M | 350M | 1.3B | 2.7B | 6.7B | 13B | 7B | 13B |
| Magnitude | AbsMax | 1s | 1s | 1s | 1s | 2s | 4s | 2s | 4s |
| SparseGPT | OPTQ | 1m | 2m | 5m | 11m | 22m | 41m | 25m | 46m |
| Wanda | SLɪM-Quant | 0.5m | 1m | 3m | 5m | 8m | 13m | 8m | 14m |
| Wanda-SVD | SLɪM-Quant | 1m | 2m | 7m | 13m | 33m | 60m | 38m | 67m |
| SLɪM | SLɪM-Quant | 1m | 2m | 7m | 13m | 34m | 63m | 39m | 68m |

Table 10: The inference time of feed forward layers in LLaMA-2-13B for different weight sizes. The speedup is computed as $\frac{\text{Dense Time}}{\text{Sparse Time}+\text{Low-rank Time}}$, and the quantization is disabled in this case.

| Matrix | Dense Time | Sparse Time | Low-rank Time | Speedup |
|---|---|---|---|---|
| $Q, K, V, O_{proj}$ | 0.96ms | 0.44ms | 0.22ms | 1.46× |
| Upsample | 2.33ms | 1.45ms | 0.39ms | 1.27× |
| Downsample | 2.19ms | 1.22ms | 0.37ms | 1.37× |

We additionally conduct an experiment with 8-bit weight quantization and add the low-rank adapters with rank ratio of 0.1 to each feedforward layer of the model. Figure ?? summarizes the layer-wise speedups achieved for different weight sizes used in LLaMA and OPT models. The baselines are for dense models with float-16 data types and all these results are obtained on NVIDIA RTX-3060 GPUs. We use a batch size of 8 to mimic the use of LLMs at inference time, when different tokens are generated.

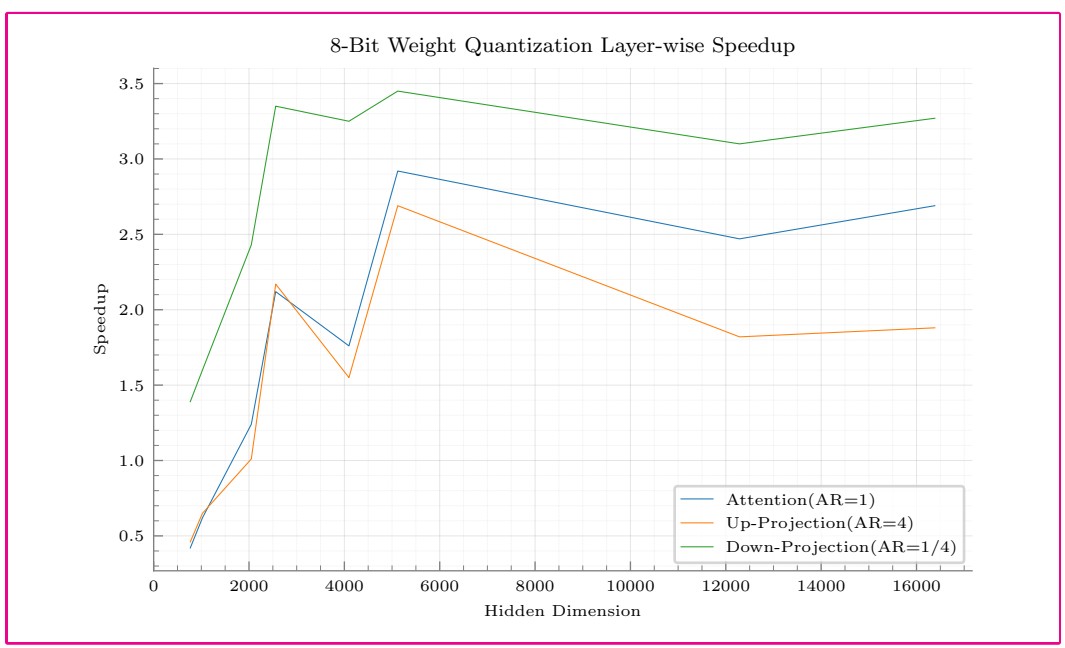

Figure 2: 8-bit weight quantization speedup (×) for up-projection, down-projection, and attention layers of different models. AR stands for the aspect ratio of the weight matrix, and is defined as $\frac{d_{out}}{d_{in}}$. ↑ indicates better performance.

 The results show that adding low-rank adapters do not hinder the speedup achieved by quantization, except for some aspect ratios with the hidden dimensions less than 1024, which correspond to model sizes of less than 125M parameters. In most practical cases with models with billions of parameters, the quantization with the low-rank adapters leads to speedups.

Please note that this shows a lower bound on the speedup that SLiM can achieve, and adding 2:4 sparsity and reducing the quantization bitwidth to 4 bits will increase the achieved speedup even further.

## H    RANK ANALYSIS

The key hyperparameter in low-rank approximation is the rank of the adapters. While increasing the rank reduces approximation error, it also leads to higher computational and memory overhead. Therefore, it is crucial to analyze the trade-off between the accuracy improvements and the overhead introduced by the chosen approximation rank.

Assuming the rank of the low-rank adapter is $rd$, where $r < 1$ is a fixed factor and $d$ is the dimension of the weights in a square feed-forward layer, the low-rank adapters are represented as $\mathcal{L}, \mathcal{R}^T \in \mathbb{R}^{d \times rd}$, resulting in a memory overhead of $\mathcal{O}(2rd^2)$ for storing them. To compute $\mathcal{X}\mathcal{L}\mathcal{R}$, where $\mathcal{X} \in \mathbb{R}^{b \times d}$ is the input with a batch size of $b$, the computational complexity is $\mathcal{O}(2brd^2)$. Given that the original memory and computational complexity of the layer are $\mathcal{O}(d^2)$ and $\mathcal{O}(bd^2)$, respectively, the overhead introduced by the low-rank adapters becomes negligible when $r \ll 1$.

Figure H-a shows the average zero-shot accuracy of the OPT-6.7B and LLaMA-2-7B models for various ranks. As expected, increasing the rank leads to improved model accuracy. Based on these results, a rank of $r = 0.1$ provides a substantial boost in accuracy without introducing significant overhead to inference.

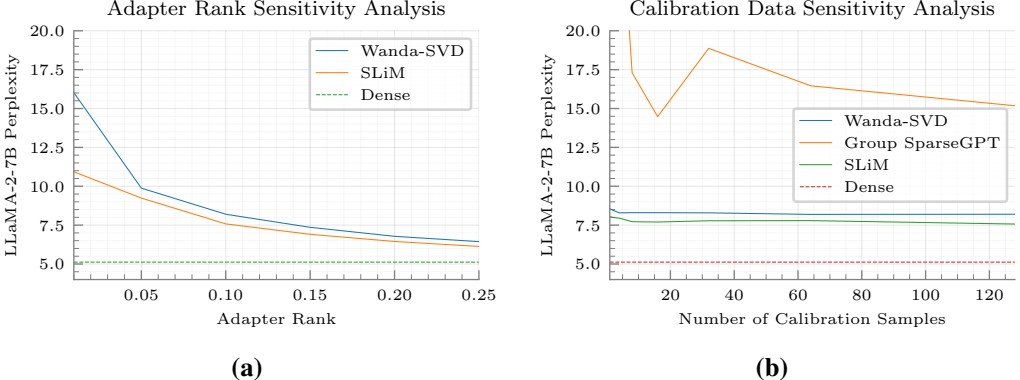

(a)                                                      (b)

Figure 3: Sensitivity analysis for the rank of the adapter (**a**) and the number of calibration samples (**b**) for different one-shot compression methods. For Wanda-SVD and SLiM , we have used the SLiM-Quant quantization method, and for the SparseGPT, we have used the Group quantization version of OPTQ. The non-group version of OPTQ and Wanda without low-rank adapters lead to divergence, and hence are not included in this figure.

## I    EFFECTS OF CALIBRATION SAMPLE COUNT

Similar to SparseGPT and Wanda, SLiM leverages a set of calibration data from the C4 dataset to assess weight saliency for pruning and low-rank approximations. Figure H-b illustrates the perplexity of LLaMA-2-7B using varying numbers of calibration samples. As shown, SLiM demonstrates low sensitivity to the number of calibration samples, making it effective even in scenarios with limited data.

