# OpenReview forum: "SLiM: One-shot Quantized Sparse Plus Low-rank Approximation of LLMs"
_ICLR.cc/2025/Conference — Submitted to ICLR 2025_

### Official Review · Reviewer_i42N · 2024-10-29

**Soundness:** 2
**Presentation:** 2
**Contribution:** 1
**Rating:** 3
**Confidence:** 5

**Summary:**

The paper introduces SLIM, a method for compressing Large Language Models (LLMs) using a one-shot approach that combines quantization, sparsity, and low-rank approximation. It reduces memory usage and inference time without retraining, while maintaining high accuracy. SLIM also includes an efficient fine-tuning process, achieving up to 5.8% accuracy improvement over existing compression techniques, making it ideal for deploying large models on resource-constrained hardware.

**Strengths:**

1. The proposed symmetric quantization minimizes quantization error without altering the weights, resulting in reduced computational and memory overhead.
2. SLIM introduces a saliency-based method to reconstruct weights based on their importance, targeting weights with the highest impact on model output, which helps retain model accuracy.
3. SLIM improves model accuracy by up to 5.4% over state-of-the-art pruning and quantization methods for 2:4 sparsity patterns. With fine-tuning, the accuracy gain can increase to 5.8%.
4. The fine-tuning recipe for sparse, quantized models significantly reduces the time and resources needed for fine-tuning. For example, a 13B parameter model that typically takes 36 days to fine-tune is reduced to just 14 hours on an H100 GPU.

**Weaknesses:**

1. The specific method for determining the quantization scale in L199-L202 should be described in more detail to clarify the contribution. Some background information (e.g., symmetric quantization, description of quantization errors, etc.) could be shortened or moved to the supplementary material to reduce the length.
2. Some more advanced quantization methods [1-5] significantly outperform GPTQ. Would combining these methods further enhance compression performance?
3. The combination of these different compression methods (sparsity, quantization) should result in a higher compression rate. However, related experiments on the memory footprint of the compressed model and the analysis of inference efficiency are not included in the paper.


[1] SmoothQuant: Accurate and Efficient Post-Training Quantization for Large Language Models. ICML 2023.
[2] AWQ: Activation-aware Weight Quantization for LLM Compression and Acceleration. MLSys 2024.
[3] OmniQuant: Omnidirectionally Calibrated Quantization for Large Language Models. ICLR 2024.
[2] AffineQuant: Affine Transformation Quantization for Large Language Models. ICLR 2024.
[3] QuaRot: Outlier-Free 4-Bit Inference in Rotated LLMs. Arxiv 2024.

**Questions:**

1. The issues mentioned in the weaknesses should be addressed.
2. In L205-206, why can the impact of weight matrix quantization and pruning be modeled as additive noise? Is there any experimental or theoretical support for this claim? Why wouldn’t the combination of the two errors introduce larger errors? Intuitively, if $W^Q= W+ E_Q$ and $W^S =W+ E_S $, why would $W^Q+W^S=W_C$?

---

> ### Author Response · Authors · 2024-11-21
> **Clarifications on Quantization Methodology, Memory and Compute Efficiency, and Performance of Advanced Quantization Methods**
>
> # Part 1 / 3
>
>
> Thank you for your insightful feedback and for identifying areas that require more clarity and evaluation. We have addressed each of your concerns in the responses below.
>
> ---
>
> # Clarification on Quantization Scale Determination
>
> We apologize for the lack of clarity regarding the quantization method. To address your concern, Algorithm 1 summarizes how the multi-grid approach in SLiM-Quant works. To minimize the objective function in Equation 7, we compute the integrals in Equations 5 and 6 for different values of $\alpha$, uniformly distributed between 0 and the largest element in the weight matrix. We then narrow down the search near the regions where the objective function is smaller, performing a more accurate search by varying $\alpha$ with a smaller step size to identify the optimal quantization parameter. We will revise the paper to include this detailed explanation, and consider moving some of the background information (e.g., symmetric quantization and description of quantization errors) to the supplementary material to improve the readability of the main text.
>
>
> ---
>
>
>
> # Memory Footprint and Inference Efficiency
>
> Per your suggestion, we have provided an analysis of the memory reduction and compute reduction (similar to JSQ) of our work in the following tables.
>
> ## Memory Reduction vs. Dense Baseline (Lower is Better):
>
> | Model                          | OPT 125M | OPT 350M | OPT 1.3B | OPT 2.7B | OPT 6.7B | OPT 13B | LLaMA 2 7B | LLaMA 2 13B |
> |--------------------------------|----------|----------|----------|----------|----------|----------|-------------|--------------|
> | SparseGPT + OPTQ              | 0.40      | 0.30      | 0.25     | 0.17     | 0.15     | 0.14     | 0.15        | 0.14         |
> | Wanda + AbsMax                | 0.40      | 0.30      | 0.25     | 0.17     | 0.15     | 0.14     | 0.15        | 0.14         |
> | Wanda-SVD + AbsMax            | 0.5      | 0.42     | 0.38     | 0.31     | 0.3      | 0.29     | 0.31        | 0.30          |
> | SLiM + SLiM-Quant             | 0.50      | 0.42     | 0.38     | 0.31     | 0.30      | 0.29     | 0.31        | 0.30          |
> | Quantized SLiM + SLiM-Quant   | 0.42     | 0.33     | 0.28     | 0.20      | 0.19     | 0.18     | 0.19        | 0.18         |
>
> ---
>
> ## Compute (FLOP) Reduction vs. Dense Baseline (Higher is Better):
>
> | Method                        | OPT 125M | OPT 350M | OPT 1.3B | OPT 2.7B | OPT 6.7B | OPT 13B | LLaMA 2 7B | LLaMA 2 13B |
> |------------------------------ |----------|----------|----------|----------|----------|----------|-------------|--------------|
> | SparseGPT + OPTQ             | 1.52     | 1.66     | 1.75     | 1.91     | 1.94     | 1.96     | 1.95        | 1.97         |
> | Wanda + AbsMax               | 1.52     | 1.66     | 1.75     | 1.91     | 1.94     | 1.96     | 1.95        | 1.97         |
> | Wanda-SVD + AbsMax           | 1.32     | 1.39     | 1.43     | 1.50     | 1.51     | 1.52     | 1.49        | 1.49         |
> | SLiM + SLiM-Quant            | 1.32     | 1.39     | 1.43     | 1.50     | 1.51     | 1.52     | 1.49        | 1.49         |
> | Quantized SLiM + SLiM-Quant  | 1.32     | 1.39     | 1.43     | 1.50     | 1.51     | 1.52     | 1.49        | 1.49         |
>
> ---
>
> As you can see in the above tables, SLiM with quantized adapters (Quantized SLiM + SLiM-Quant) provides comparable memory and compute reductions to other methods without low-rank adapters, showing that the cost of additional adapters is not significant.
>
> It is noteworthy that, as mentioned in the paper and to the best of our knowledge, there is currently no open-source implementation of Int-4 x Int-8 quantization CUDA code available. Implementing such kernels is out of the scope of this paper and is planned for future work.
>
> We have included these analysis and discussions in Appendix-C and Appendix-D in pages 15 and 16 in our revised manuscript.
>
> ---

---

> > ### Author Response · Authors · 2024-11-21
> > **Clarifications on Quantization Methodology, Memory and Compute Efficiency, and Performance of Advanced Quantization Methods**
> >
> > # Part 2 / 3
> >
> > # Performance of Advanced Quantization Methods
> >
> > We appreciate your suggestion. In response, we have tested additional quantization methods you suggested (OmniQuant [1], AWQ [2], AffineQuant [3]), including the additional method JSQ [4], and provided the perplexity results on the WikiText2 dataset. Since these quantization methods do not have a pruning strategy designed for them, we pruned the models using Wanda and then applied the respective quantization methods. We excluded results for SmoothQuant [5] as it does not support 4-bit weight quantization in their method. We are currently working on reproducing the results from QuaRot [6] as well, and will report their results as soon as we get them. In all these experiments (except for JSQ), the weight is pruned with Wanda and quantized to 4 bits. The inputs are quantized to 8 bits as well. The optimal parameters for each model are used if the papers reported them, otherwise, the default quantization parameters are used.
> >
> > ## 2:4 Sparsity
> >
> > | Pruning Method          | Weight Quantization | OPT-125M | OPT-350M | OPT-1.3B | OPT-2.7B | OPT-6.7B | OPT-13B | LLaMA-2-7B | LLaMA-2-13B |
> > |---|--|--|---|--|--|----|---|----|-----|
> > | Dense                   | -                   | 27.66    | 22.00    | 14.62    | 12.47    | 10.86    | 10.13    | 5.12        | 4.57         |
> > | JSQ | JSQ         | 3.21e3   | 1.97e4   | 71.97    | 2.87e3   | 23.76    | 596.60   | 11.35       | 8.25         |
> > | Wanda                   | OmniQuant           | 97.12    | NaN      | 34.86    | 27.03    | 18.91    | 18.06    | 12.78       | 6.00         |
> > | Wanda                   | AWQ                 | 83.02    | 7.30e5   | 27.88    | 21.60    | 17.93    | 16.72    | 12.79       | OOM          |
> > | Wanda                   | AffineQuant         | 86.58    | NaN      | 28.30    | 21.99    | 16.62    | 16.83    | 6.59e3      | 6.42e3       |
> > | SLiM                    | SLiM-Quant          | 58.10    | 51.70    | 20.00    | 15.80    | 12.80    | 12.10    | 7.60        | 6.60         |
> > | SLiM + FT               | SLiM-Quant          | 41.50    | 36.30    | 17.60    | 14.70    | 12.40    | 12.10    | 6.40        | 5.80         |
> >
> > ---
> >
> > ## Unstructured Sparsity
> >
> > | Pruning Method | Weight Quantization | OPT-125M | OPT-350M | OPT-1.3B | OPT-2.7B | OPT-6.7B | OPT-13B | LLaMA-2-7B | LLaMA-2-13B |
> > |---|--|--|---|--|--|----|---|----|-----|
> > | Dense          | -                  | 27.66    | 22.00    | 14.62    | 12.47    | 10.86    | 10.13    | 5.12        | 4.57         |
> > | JSQ            | JSQ                | 3.07e3   | 2.99e4   | 24.91    | 250.99   | 15.44    | 2.30e5   | 6.66        | 5.69         |
> > | Wanda          | OmniQuant          | 44.90    | NaN      | 20.59    | 15.70    | 13.12    | 13.89    | 7.39        | 6.36         |
> > | Wanda          | AWQ                | 39.23    | 3.41e5   | 17.97    | 14.16    | 11.89    | 12.34    | 7.28        | OOM          |
> > | Wanda          | AffineQuant        | 40.91    | NaN      | 18.53    | 14.55    | 12.18    | 12.50    | 1.08e4      | 7.69e3       |
> > | SLiM           | SLiM-Quant         | 39.70    | 32.00    | 16.70    | 13.70    | 11.50    | 10.80    | 6.20        | 5.40         |
> > | SLiM + FT      | SLiM-Quant         | 34.00    | 28.30    | 15.90    | 13.40    | 11.50    | 11.50    | 5.80        | 5.20         |
> >
> >
> > As shown in the results, all the advanced quantization methods resulted in higher errors compared to SLiM (and even SparseGPT + OPTQ), as they were not co-designed with the pruning method. This highlights the importance of co-designing pruning, quantization, and low-rank adapters to achieve optimal performance.
> >
> > Furthermore, methods like JSQ demonstrate strong performance in only one class of models but struggle to generalize to others, such as the OPT models. Similarly, techniques like OmniQuant, AffineQuant, and AWQ fail to effectively quantize the OPT-350M model, often resulting in NaN (Not a Number) or extremely high perplexity values. Moreover, AffineQuant faces challenges quantizing the sparse version of LLaMA-2-7B and LLaMA-2-13B models due to the large recovery loss in the last two layers of the models. All experiments were conducted on a single A100-40GB GPU; notably, AWQ encountered out-of-memory (OOM) issues when applied to the LLaMA-2-13B model, as indicated in the table.
> >
> > It is noteworthy that none of the quantization methods we tested can be combined with saliency-based low-rank adapter methods such as SLiM or Wanda-SVD. This is because these quantization methods alter the values of the weights during the quantization process, which disrupts the saliency-based optimization of low-rank adapters. This further emphasizes the importance of co-designing pruning, quantization, and low-rank adapters to ensure compatibility and maintain optimal performance.
> >
> > We have included these experiments and discussion in Appendix-B-Page-14 and Table 6 in Page-16 in our revised manuscript.
> >
> > ---

---

> > > ### Author Response · Authors · 2024-11-21
> > > **Clarifications on Quantization Methodology, Memory and Compute Efficiency, and Performance of Advanced Quantization Methods**
> > >
> > > # Part 3 / 3
> > >
> > > # Modeling the Impact of Weight Matrix Quantization and Pruning as Additive Noise
> > >
> > > The quantization error is defined as $E_Q = W - W^Q$ and the pruning error is defined as $E_S = W^C - W^Q$, where $W$ is the original weight matrix, $W^Q$ is the quantized weight matrix, and $W^C$ is the pruned and quantized weight matrix. As a result, these errors can be modeled as additive noise (since they are added to the original/quantized matrices). The combination of these errors will result in $W^C = W - E_Q - E_S$. Thus, the final noise is the sum of the quantization and pruning errors. While the norm of this combined error can be larger than $E_Q$ and $E_S$ individually, this additive property is utilized in the computation of SLiM low-rank adapters.
> > > We have included this discussion in Page 4 of our revised manuscript.
> > >
> > > ---
> > >
> > > Once again, we are grateful for your detailed feedback. We hope our responses have effectively resolved your concerns and support a potential re-assessment of the score. Should you have any further questions, we would be happy to discuss them during the designated period. Thank you for your time and consideration.
> > >
> > > ---
> > >
> > > [1] Shao, et al. “OmniQuant: Omnidirectionally Calibrated Quantization for Large Language Models.” ICLR 2024.
> > >
> > > [2] Lin, et al. “AWQ: Activation-aware Weight Quantization for On-Device LLM Compression and Acceleration.” MLSys 2024
> > >
> > > [3] Ma, et al. “AffineQuant: Affine Transformation Quantization for Large Language Models.” ICLR 2024.
> > >
> > > [4] Guo, et al. “Compressing large language models by joint sparsification and quantization.” ICML 2024
> > >
> > > [5] Xiao, et al. “SmoothQuant: Accurate and Efficient Post-Training Quantization for Large Language Models.” ICML 2023
> > >
> > > [6] Ashkboos, et al. “QuaRot: Outlier-Free 4-Bit Inference in Rotated LLMs.” NeurIPS 2024

---

> > > > ### Comment · Reviewer_i42N · 2024-11-27
> > > > **I still have significant concerns about this work.**
> > > >
> > > > Thanks for the authors' detailed response, but I still have significant concerns about this work. The theoretical foundation is particularly problematic - especially in Section 3.3, where they simply assumed quantization and pruning errors can be combined without proper justification. The saliency function is directly borrowed from previous work without proving its applicability in this context. The whole theoretical framework feels like it was reverse-engineered from experimental results rather than being rigorously developed.
> > > >
> > > > While they provided comprehensive experimental results, there are still several issues that weren't adequately addressed. They acknowledge the lack of implementation supporting both 2:4 sparsity and 4-bit quantization, which limits the practical value of their method. Some baselines show abnormal results (like NaN or extremely high perplexity on OPT-350M) without proper analysis, and results from important comparison methods like SmoothQuant and QuaRot are still missing.
> > > >
> > > > The method's novelty is also questionable, as it mainly combines existing approaches without substantial theoretical innovation. The engineering implementation has significant limitations, and the practical benefits are not clearly demonstrated. Given these concerns, I maintain my original rating (3: reject). I suggest the authors focus on strengthening the theoretical foundation and improving the implementation to make the method more practical and theoretically sound.

---

### Official Review · Reviewer_7uwS · 2024-11-03

**Soundness:** 2
**Presentation:** 2
**Contribution:** 3
**Rating:** 5
**Confidence:** 4

**Summary:**

In this paper, the authors propose to combine quantization and pruning to compress LLMs. A new quantization method called SLiM-Quant is proposed to reduce the quantization error. To develop the new method, the mean squared error used in the literature was converted to a probabilistic formulation, which is solved by numerical integration over the weight histogram. Then a pruning method (structured or unstructured pruning) is applied. To compensate for the accuracy loss due to pruning and quantization, a new low-rank adapter is proposed, which is based on the assumption of an additive invertible saliency function. Finally, the low-rank adapters are quantized in a manner of 16x16 tiles, which is finally merged into the quantized sparse weight.

**Strengths:**

1. Experimental results show that the proposed method leads to improved model accuracy under the same pruning and quantization scheme.

2. A novel low-rank adapter is proposed in this paper.

**Weaknesses:**

1. But why is the proposed low-rank adapter not compared with the previous low-rank adapter? Without such a comparison, it's hard to tell the advantage of the proposed method.

2. The combination of quantization and pruning seems to be a straightforward method. In this paper, the two methods seems to be loosely coupled. As mentioned in Questions, the joint optimization has some unanswered questions.

3. Actually acceleration performance is not reported in this paper.

**Questions:**

1. When merging the quantized sparse weight and low-rank adapters, there might be overflow due to addition. The non-zero values from the low-rank adapters might also be added to originally the pruned weights, which disturbs the pruning mechanism (in particular the 2:4 structured pruning). How should those potential problems during weight merging be solved?

2. What is the evaluation metric used in Table 1-3?

3. Does the quantization method proposed in this paper lead to zero values?

4. The paper did not propose a new pruning method. But in Table, SLiM works better than other methods under a pruning-only scheme.

5. The additive invertible saliency function comes from nowhere. The property is not explained in the supplementary either.

---

> ### Author Response · Authors · 2024-11-21
> **Clarification on Low-Rank Adapter Methods, Acceleration Performance, Quantization Effects, and Evaluation Metrics**
>
> # Part 1 / 2
>
> We appreciate the reviewer’s detailed feedback and suggestions for improvement. In the following response, we address all your questions and requests for additional evaluation.
>
> ---
>
> # Justifying the Comparison with Previous Low-Rank Adapter Methods
>
> Thank you for your comment. Previous works in low-rank adapters either use zero initialization (LoRA [1], Q-LoRA [2]) or use SVD for initialization (LQ-LoRA [3], LoSparse [4]). The most relevant prior work in the field of low-rank adapters is LQ-LoRA, which uses SVD for low-rank initialization and does not incorporate sparsity. In contrast, our work introduces sparsity alongside low-rank adapters. To provide a fair comparison, we implemented a variant of LQ-LoRA by pruning the weights using the Wanda method, which we refer to as Wanda-SVD in our paper. This allows us to compare the impact of adding sparsity to LQ-LoRA.
>
> As shown in Table 1 in our manuscript, our approach, SLiM, consistently outperforms LQ-LoRA (Wanda-SVD) across various benchmarks. This demonstrates the advantage of combining sparsity with low-rank adapters, highlighting the effectiveness of our method.
>
>
> ---
>
> # Providing Acceleration Performance Metrics
>
> Thank you for your comment. As per your request, we have included the analysis of compute reduction (in terms of FLOPs) and memory reduction for our method in the following tables.
>
> The tables below demonstrate the performance in terms of memory and compute reductions, comparing SLiM (with and without quantized adapters) against other methods. As shown, SLiM provides competitive memory and compute reductions similar to other methods that do not use low-rank adapters, illustrating that the cost of additional adapters is minimal.
>
> **Memory reduction vs. the dense baseline (lower is better):**
>
> | Model                          | OPT 125M | OPT 350M | OPT 1.3B | OPT 2.7B | OPT 6.7B | OPT 13B | LLaMA 2 7B | LLaMA 2 13B |
> |---------------------------------|----------|----------|----------|----------|----------|----------|-------------|-------------|
> | SparseGPT + OPTQ               | 0.40      | 0.30      | 0.25     | 0.17     | 0.15     | 0.14     | 0.15        | 0.14        |
> | Wanda + AbsMax                 | 0.40      | 0.30      | 0.25     | 0.17     | 0.15     | 0.14     | 0.15        | 0.14        |
> | Wanda-SVD + AbsMax             | 0.50      | 0.42     | 0.38     | 0.31     | 0.30      | 0.29     | 0.31        | 0.30         |
> | SLiM + SLiM-Quant              | 0.05      | 0.42     | 0.38     | 0.31     | 0.03      | 0.29     | 0.31        | 0.30         |
> | Quantized SLiM + SLiM-Quant    | 0.42     | 0.33     | 0.28     | 0.20      | 0.19     | 0.18     | 0.19        | 0.18        |
>
> **Compute (FLOP) reduction vs. the dense baseline (higher is better):**
>
> | Method                        | OPT 125M | OPT 350M | OPT 1.3B | OPT 2.7B | OPT 6.7B | OPT 13B | LLaMA 2 7B | LLaMA 2 13B |
> |-------------------------------|----------|----------|----------|----------|----------|----------|-------------|-------------|
> | SparseGPT + OPTQ               | 1.52     | 1.66     | 1.75     | 1.91     | 1.94     | 1.96     | 1.95        | 1.97        |
> | Wanda + AbsMax                 | 1.52     | 1.66     | 1.75     | 1.91     | 1.94     | 1.96     | 1.95        | 1.97        |
> | Wanda-SVD + AbsMax             | 1.32     | 1.39     | 1.43     | 1.50     | 1.51     | 1.52     | 1.49        | 1.49        |
> | SLiM + SLiM-Quant              | 1.32     | 1.39     | 1.43     | 1.50     | 1.51     | 1.52     | 1.49        | 1.49        |
> | Quantized SLiM + SLiM-Quant    | 1.32     | 1.39     | 1.43     | 1.50     | 1.51     | 1.52     | 1.49        | 1.49        |
>
> As seen, **Quantized SLiM + SLiM-Quant** offers memory and compute reductions comparable to other methods without low-rank adapters, which confirms that the additional cost of low-rank adapters is negligible in terms of performance.
>
> It is important to note that, as mentioned in the paper, currently there is no open-source implementation of Int-4 x Int-8 sparse matrix-matrix multiplication CUDA code available, and implementing such kernels is out of the scope of this paper. We have outlined this as part of our future work.
>
> ---

---

> > ### Author Response · Authors · 2024-11-21
> > **Clarification on Low-Rank Adapter Methods, Acceleration Performance, Quantization Effects, and Evaluation Metrics**
> >
> > # Part 2 / 2
> >
> > # Addressing Potential Overflow and Disturbance from Low-Rank Adapter Merging
> >
> > Thank you for raising this important point. In SLiM, we do not merge the low-rank adapters with the original weights. This design decision ensures that the original weights remain sparse and quantized, preventing issues like overflow during merging or disturbance to the pruning mechanism, particularly for methods like 2:4 structured pruning. By maintaining separate handling for the adapters and the pruned weights, we can preserve the integrity of both the sparsity and low-rank structure.
> >
> > As a result, concerns regarding the addition of non-zero values from the low-rank adapters to the originally pruned weights, which might interfere with the pruning process, do not arise in SLiM. Furthermore, the memory and compute overhead of incorporating low-rank adapters is minimal compared to the performance gains achieved, as shown in the memory and compute reduction tables provided earlier.
> >
> > ---
> >
> > # Clarifying Evaluation Metrics for Tables 1-3
> >
> > Thank you for your question. In Tables 1-3, we report the average accuracy of the models over several zero-shot downstream tasks, including MMLU, Piqa, Arc-Easy, Arc-Challenge, WinoGrande, and OpenBookQA. These tasks evaluate the performance of the models across a range of reasoning and question-answering challenges.
> >
> > More detailed information about these datasets and the corresponding experimental setup can be found in Lines 315 to 323 of the paper.
> >
> > ---
> >
> >
> > # Quantization Leading to Zero Values
> >
> > The quantization method proposed in this paper is symmetric. This means that values between $-\frac{\Delta}{2}$ and $\frac{\Delta}{2}$, where $\Delta$ is the quantization step, will be mapped to zero. While this introduces additional zeros in the layer, it does not negatively affect the accuracy of the model, as demonstrated in our results.
> >
> > ---
> >
> > # Clarification on Pruning Method
> >
> > We apologize for the lack of clarity regarding our pruning method. As mentioned in the paper, we use the Wanda saliency metric for pruning, meaning that our pruning method is similar to Wanda. However, SLiM goes beyond pruning by incorporating low-rank adapters into the model using a saliency-based approach, which enhances the model's accuracy. This additional step is why SLiM performs better than other methods in the pruning-only scheme.
> >
> > We have included this discussion in Page 7 of our revised manuscript.
> >
> > ---
> >
> >
> > # Clarification on Additive Invertible Saliency Function
> >
> > We apologize for the confusion regarding the additive invertible saliency function. By "additive saliency," we mean that for two arbitrary matrices $A$ and $B$, the saliency function $F$ satisfies $F(A + B) = F(A) + F(B)$. This property enables us to compute the saliency of additive errors separately, which allows us to optimize the low-rank adapters effectively to minimize these errors. In contrast, other saliency methods, such as the Hessian inverse used in GPTQ or OBS [5], do not exhibit this additive property. As a result, saliency-based low-rank adapters cannot be applied in the same way to those methods.
> >
> > We have included this discussion in Page 6 of our revised manuscript.
> >
> > ---
> >
> > We sincerely thank you for your valuable feedback. We hope our responses have sufficiently addressed your concerns and warrant a reconsideration of the score. We are open to further discussion and would be glad to answer any additional questions during the discussion period. Your time and thoughtful review are greatly appreciated.
> >
> > ---
> >
> > [1] Hu, et al. “LoRA: Low-Rank Adaptation of Large Language Models.”, ICLR 2022
> >
> > [2] Dettmers, et al. “QLoRA: Efficient Finetuning of Quantized LLMs.”, NeurIPS 2023
> >
> > [3] Guo, et al. “LQ-LoRA: Low-rank Plus Quantized Matrix Decomposition for Efficient Language Model Finetuning.” ICLR 2024
> >
> > [4] Li, et al. “LoSparse: Structured Compression of Large Language Models based on Low-Rank and Sparse Approximation.” ICML 2023

---

> > > ### Comment · Reviewer_7uwS · 2024-11-26
> > >
> > > After reading the authors' rebuttal, I'm still concerned with the actual acceleration of the proposed method. The authors claimed that the weights and low-rank adapters are not merged. This means that two separated linear operations should be conducted with respect to the weights and low-rank adapters. I guess this is the main reason why the authors only show the computation and memory footprint instead of actual acceleration in the rebuttal.  Thus, I will keep my original rating of this paper.

---

> > > > ### Author Response · Authors · 2024-11-27
> > > > **Additional Speedup Experiments**
> > > >
> > > > Dear Reviewer 7uwS,
> > > >
> > > > We greatly appreciate your detailed review and thoughtful feedback. To address your concerns, we would like to clarify and further elaborate on the speedup analysis for our proposed method.
> > > >
> > > > To the best of our knowledge, there is currently no open-source repository that supports both 2:4 sparsity and the 4-bit quantization approach described in our manuscript. Consequently, we have limited our speedup experiments to scenarios involving either sparsity or quantization, for which efficient implementations are readily available. Developing optimized kernels for combined sparsity and quantization is indeed beyond the scope of this work, and we consider it a promising direction for future research.
> > > >
> > > > In our submission, we included the layer-wise speedup results of pruning methods (without quantization), as reported in Table 10 (Page 18) of the revised manuscript. These results demonstrate that the computational overhead of incorporating low-rank adapters is negligible compared to the forward-pass time of both sparse and dense feedforward layers.
> > > >
> > > > To further address your concerns, we conducted an additional experiment to provide a **lower bound** on the achievable speedup with SLiM. In this experiment, we applied 8-bit weight quantization and introduced low-rank adapters with a rank ratio of 0.1 to each feedforward layer of the model. The results, summarized in the table below, compare layer-wise speedups for various weight sizes used in LLaMA and OPT models. These experiments were conducted on NVIDIA RTX-3060 GPUs with a batch size of 8, simulating the token generation phase of inference in large language models.
> > > >
> > > > ### 8-Bit Weight Quantization Speedup ($\times$)
> > > >
> > > > AR denotes the aspect ratio (output dimension / input dimension) of the layer. $\uparrow$ indicates better performance.
> > > >
> > > > | Hidden Dimension       | 768   | 1024  | 2048  | 2560  | 4096  | 5120  | 12288 | 16384 |
> > > > |-------------------------|-------|-------|-------|-------|-------|-------|-------|-------|
> > > > | Down-Projection (AR=1/4) | 1.39  | 1.6   | 2.43  | 3.35  | 3.25  | 3.45  | 3.1   | 3.27  |
> > > > | Up-Projection (AR=4)     | 0.46  | 0.65  | 1.01  | 2.17  | 1.55  | 2.69  | 1.82  | 1.88  |
> > > > | Attention Layers (AR=1)  | 0.42  | 0.62  | 1.24  | 2.12  | 1.76  | 2.92  | 2.47  | 2.69  |
> > > >
> > > > Figure 2 on Page 18 of the revised manuscript visualizes these results. They reveal that adding low-rank adapters does not significantly hinder the speedup achieved by quantization, except for certain aspect ratios with hidden dimensions smaller than 1024 (corresponding to model sizes below 125M parameters). Importantly, for larger models—those with billions of parameters, which are more relevant in practice—the combination of quantization and low-rank adapters yields meaningful speedups.
> > > >
> > > > We would also like to highlight that these results represent a **lower bound** on the potential speedup achievable with SLiM. Incorporating 2:4 sparsity and further reducing the quantization bitwidth to 4 bits would likely enhance these speedups even further.
> > > >
> > > > We have included this discussion in Appendix G (Page 18) of our revised draft. We hope this additional experiment and clarification address your concerns. Once again, we thank you for your valuable feedback and for helping us strengthen our work.

---

### Official Review · Reviewer_6F8M · 2024-11-04

**Soundness:** 2
**Presentation:** 2
**Contribution:** 2
**Rating:** 3
**Confidence:** 4

**Summary:**

The manuscript proposes an LLM compression method that combines quantization, sparsity, and low-rank approximation. This approach applies low-rank approximation guided by a significance-driven optimization function, followed by fine-tuning of LoRA blocks, ultimately achieving advanced improvements in compression accuracy.

**Strengths:**

1. Previous algorithms aimed at compression during inference generally do not retain LoRA. This manuscript innovatively retains it, and demonstrates its feasibility in terms of time efficiency through inference efficiency analysis.

2. The idea of using saliency to guide the optimization of LoRA is highly novel.

**Weaknesses:**

1. The motivation for introducing Equation 8 is unclear. For instance, the authors claim that deriving LR from E_Q and E_S is intended to “cancel the compression errors”, yet introduce a significance function F that does not seem directly related to these compression errors.

2. Prior to Equation 10, the significance calculation method from Wanda is introduced without clear motivation. To my knowledge, there are alternative methods for measuring significance, such as AWQ[1] and SparseGPT[2], yet the authors appear not to consider them.

3. In Section 3.2, the manuscript claims that previous error minimization methods did not consider "the importance of different weights", though methods like OBS[3] clearly address the importance of weights (see its explanation of L_q).

4. The manuscript does not explain the pruning approach used and lacks relevant formula descriptions for E_S.

5. There is a lack of comparison with advanced joint strategies for sparse quantization, such as JSQ[4].

6. The manuscript lacks statistics on model size, which may lead to unfair comparisons. Since SLIM introduces additional LoRA layers with a substantial rank (r=0.1), describing the extra parameter overhead is necessary. Similarly, theoretical computations for the number of OPs are also not provided.

[1] Lin, et al. AWQ: Activation-aware Weight Quantization for On-Device LLM Compression and Acceleration[J].

[2] Frantar, et al. Sparsegpt: Massive language models can be accurately pruned in one-shot.

[3] Hassibi, et al. Optimal brain surgeon and general network pruning.

[4] Guo, et al. Compressing large language models by joint sparsification and quantization.

**Questions:**

1. Significance is usually used to guide pruning decisions, so why is the significance calculation method proposed in Wanda used as the optimization target for quantization and sparsification? Why does optimizing the significance metric result in better low-rank adapters?

2. Why does not pruning weights in LQ-LORA lead to a significantly higher error rate?

3. Why does the discussion on Inference Speedup not consider the impact of quantization?

---

> ### Author Response · Authors · 2024-11-21
> **Motivation of Equation 8, Wanda’s Additive Property, Low-Rank Adapter Significance, Advanced Joint Quantization, and Computational Overhead**
>
> # Part 1 / 4
>
>
> Thank you for your thorough review and for highlighting areas where further clarification and evaluation would be beneficial. We address each of your questions and concerns in detail below.
>
> ---
>
> # Clarifying the Motivation Behind Equation 8
>
> The motivation behind introducing Equation 8 is to leverage the additive property of the saliency function $F$. Specifically, this allows us to compute the saliency of the error components, $E_Q$ and $E_S$, and subsequently minimize the saliency of the combined compression error rather than its norm. This approach is grounded in our observation that directly minimizing the norm of the compression error can inadvertently result in higher overall errors. In contrast, minimizing the saliency-weighted error provides a more targeted reduction of critical error components, as demonstrated in our experiments. This refinement leads to improved performance, showcasing the effectiveness of our method in handling compression errors more efficiently.
>
> ---
>
> # Justifying the Choice of Wanda for Significance Calculation
>
> While there are alternative saliency methods like AWQ[1] and SparseGPT[2], we chose the Wanda saliency method due to its unique additive property, where $F(A + B) = F(A) + F(B)$. This property is crucial for our approach, as it enables us to assess the saliency of the combined compression error components and selectively recover significant parts. In contrast, other methods, such as the Hessian-based approach in SparseGPT, lack this additive nature. For instance, $(\nabla^2_{W_C + E} \mathcal{L})^{-1} \neq (\nabla^2_{W_C} \mathcal{L})^{-1} + (\nabla^2_{E} \mathcal{L})^{-1}$, which limits their ability to effectively utilize one-shot low-rank adapters. Thus, we opted for Wanda to ensure accurate saliency calculations aligned with the objectives of our method.
>
> We have included this discussion in Page 6 of our revised manuscript.
>
> ---
>
> # Clarifying the Scope of Weight Importance in Low-Rank Adapter Methods
>
> Our claim is specifically focused on **previous low-rank adapter methods** like LQ-LoRA, where the importance of individual weights is not explicitly considered. We are not suggesting that pruning and quantization methods, such as OBS[3], overlook weight importance—indeed, these methods, including the use of Wanda’s importance metric, do account for it. However, in the context of low-rank adapters, existing approaches typically focus only on minimizing the magnitude of the errors, rather than assessing the significance of different weights. Our work addresses this gap by incorporating weight importance into the error minimization process for low-rank adaptations.
>
> ---
>
> # Clarifying the Pruning Approach and Definition of Sparsity Error
>
> We apologize for the lack of clarity regarding the pruning approach. In our method, we employ the Wanda saliency technique for pruning. Specifically, after quantizing the weights to obtain $W^Q$, we apply the Wanda method on these quantized weights to derive $W^C$. The sparsity error is then defined as $E_S = W^C - W^Q$, capturing the difference introduced by pruning. This approach ensures that the pruning process is guided by the saliency of the weights, thereby preserving the most significant components for model performance. We will include additional details and relevant formulae in the revised manuscript to enhance clarity.
>
> We have included these discussions in Pages 4 and 7 of our revised manuscript.
>
> ---

---

> > ### Author Response · Authors · 2024-11-21
> > **Motivation of Equation 8, Wanda’s Additive Property, Low-Rank Adapter Significance, Advanced Joint Quantization, and Computational Overhead**
> >
> > # Part 2 / 4
> >
> > # Addressing Comparison with Advanced Joint Sparse Quantization Strategies
> >
> > Thank you for pointing out the lack of comparison with advanced joint strategies like JSQ[4]. In response to your feedback, we have conducted additional experiments to include JSQ (with its default pruning and quantization parameters) on the same benchmarks used in our paper. Furthermore, we have expanded our evaluation to incorporate other recent quantization baselines, such as AWQ[1], OmniQuant[5], and AffineQuant[6],. The updated results, including the perplexity scores of these methods on the WikiText2 dataset, are presented in the following table. This extended evaluation provides a more comprehensive comparison, highlighting the strengths of our approach against state-of-the-art techniques. In all these experiments (except for JSQ), the weight is pruned with Wanda and quantized to 4 bits. The inputs are quantized to 8 bit as well. The optimal parameters for each model are used if the papers reported them, otherwise, the default quantization parameters are used.
> >
> > ## 2:4 Sparsity
> >
> > | Pruning Method          | Weight Quantization | OPT-125M | OPT-350M | OPT-1.3B | OPT-2.7B | OPT-6.7B | OPT-13B | LLaMA-2-7B | LLaMA-2-13B |
> > |----------------------|-----------------|----------|----------|----------|----------|----------|----------|-------------|--------------|
> > | Dense                   | -                   | 27.66    | 22.00    | 14.62    | 12.47    | 10.86    | 10.13    | 5.12        | 4.57         |
> > | JSQ | JSQ         | 3.21e3   | 1.97e4   | 71.97    | 2.87e3   | 23.76    | 596.60   | 11.35       | 8.25         |
> > | Wanda                   | OmniQuant           | 97.12    | NaN      | 34.86    | 27.03    | 18.91    | 18.06    | 12.78       | 6.00         |
> > | Wanda                   | AWQ                 | 83.02    | 7.30e5   | 27.88    | 21.60    | 17.93    | 16.72    | 12.79       | OOM          |
> > | Wanda                   | AffineQuant         | 86.58    | NaN      | 28.30    | 21.99    | 16.62    | 16.83    | 6.59e3      | 6.42e3       |
> > | SLiM                    | SLiM-Quant          | 58.10    | 51.70    | 20.00    | 15.80    | 12.80    | 12.10    | 7.60        | 6.60         |
> > | SLiM + FT               | SLiM-Quant          | 41.50    | 36.30    | 17.60    | 14.70    | 12.40    | 12.10    | 6.40        | 5.80         |
> >
> > ---
> >
> > ## Unstructured Sparsity
> >
> > | Pruning Method | Weight Quantization | OPT-125M | OPT-350M | OPT-1.3B | OPT-2.7B | OPT-6.7B | OPT-13B | LLaMA-2-7B | LLaMA-2-13B |
> > |----------|----------|---------|----------|----------|----------|----------|----------|-------------|------------|
> > | Dense          | -                  | 27.66    | 22.00    | 14.62    | 12.47    | 10.86    | 10.13    | 5.12        | 4.57         |
> > | JSQ            | JSQ                | 3.07e3   | 2.99e4   | 24.91    | 250.99   | 15.44    | 2.30e5   | 6.66        | 5.69         |
> > | Wanda          | OmniQuant          | 44.90    | NaN      | 20.59    | 15.70    | 13.12    | 13.89    | 7.39        | 6.36         |
> > | Wanda          | AWQ                | 39.23    | 3.41e5   | 17.97    | 14.16    | 11.89    | 12.34    | 7.28        | OOM          |
> > | Wanda          | AffineQuant        | 40.91    | NaN      | 18.53    | 14.55    | 12.18    | 12.50    | 1.08e4      | 7.69e3       |
> > | SLiM           | SLiM-Quant         | 39.70    | 32.00    | 16.70    | 13.70    | 11.50    | 10.80    | 6.20        | 5.40         |
> > | SLiM + FT      | SLiM-Quant         | 34.00    | 28.30    | 15.90    | 13.40    | 11.50    | 11.50    | 5.80        | 5.20         |
> >
> > As shown in the results, all the advanced quantization methods resulted in higher errors compared to SLiM. Furthermore, methods like JSQ demonstrate strong performance in only one class of models but struggle to generalize to others, such as the OPT models. Similarly, techniques like OmniQuant, AffineQuant, and AWQ fail to effectively quantize the OPT-350M model, often resulting in NaN (Not a Number) or extremely high perplexity values. Moreover, AffineQuant faces challenges quantizing the sparse version of LLaMA-2-7B and LLaMA-2-13B models due to the large recovery loss in the last two layers of the models. All experiments were conducted on a single A100-40GB GPU; notably, AWQ encountered out-of-memory (OOM) issues when applied to the LLaMA-2-13B model, as indicated in the table.
> >
> > It is noteworthy that none of the quantization methods we tested can be combined with saliency-based low-rank adapter methods such as SLiM or Wanda-SVD. This is because these quantization methods alter the values of the weights during the quantization process, which disrupts the saliency-based optimization of low-rank adapters. This further emphasizes the importance of co-designing pruning, quantization, and low-rank adapters to ensure compatibility and maintain optimal performance.
> >
> > We have included these experiments and discussion in Appendix-B-Page-14 and Table 6 in Page-16 in our revised manuscript.
> >
> >
> > ---

---

> > > ### Author Response · Authors · 2024-11-21
> > > **Motivation of Equation 8, Wanda’s Additive Property, Low-Rank Adapter Significance, Advanced Joint Quantization, and Computational Overhead**
> > >
> > > # Part 3 / 4
> > >
> > > # Addressing Concerns on Model Size and Computational Overhead
> > >
> > > Thank you for highlighting the need for a detailed analysis of model size and computational overhead. To address your concerns, we have included an analysis of both the memory and compute reductions achieved by our approach, similar to the benchmarks used for JSQ. The following tables provide a comparison of memory reduction (lower is better) and compute (FLOP) reduction (higher is better) versus the dense baseline across various model sizes.
> > >
> > > ---
> > >
> > > ## Memory Reduction vs. Dense Baseline (Lower is Better)
> > >
> > > | Model                          | OPT 125M | OPT 350M | OPT 1.3B | OPT 2.7B | OPT 6.7B | OPT 13B | LLaMA 2 7B | LLaMA 2 13B |
> > > |--------------------------------|----------|----------|----------|----------|----------|----------|-------------|--------------|
> > > | SparseGPT + OPTQ               | 0.4      | 0.3      | 0.25     | 0.17     | 0.15     | 0.14     | 0.15        | 0.14         |
> > > | Wanda + AbsMax                 | 0.4      | 0.3      | 0.25     | 0.17     | 0.15     | 0.14     | 0.15        | 0.14         |
> > > | Wanda-SVD + AbsMax             | 0.5      | 0.42     | 0.38     | 0.31     | 0.3      | 0.29     | 0.31        | 0.3          |
> > > | SLiM + SLiM-Quant              | 0.5      | 0.42     | 0.38     | 0.31     | 0.3      | 0.29     | 0.31        | 0.3          |
> > > | Quantized SLiM + SLiM-Quant    | 0.42     | 0.33     | 0.28     | 0.2      | 0.19     | 0.18     | 0.19        | 0.18         |
> > >
> > > ---
> > >
> > > ## Compute (FLOP) Reduction vs. Dense Baseline (Higher is Better)
> > >
> > > | Method                         | OPT 125M | OPT 350M | OPT 1.3B | OPT 2.7B | OPT 6.7B | OPT 13B | LLaMA 2 7B | LLaMA 2 13B |
> > > |--------------------------------|----------|----------|----------|----------|----------|----------|-------------|--------------|
> > > | SparseGPT + OPTQ               | 1.52     | 1.66     | 1.75     | 1.91     | 1.94     | 1.96     | 1.95        | 1.97         |
> > > | Wanda + AbsMax                 | 1.52     | 1.66     | 1.75     | 1.91     | 1.94     | 1.96     | 1.95        | 1.97         |
> > > | Wanda-SVD + AbsMax             | 1.32     | 1.39     | 1.43     | 1.50     | 1.51     | 1.52     | 1.49        | 1.49         |
> > > | SLiM + SLiM-Quant              | 1.32     | 1.39     | 1.43     | 1.50     | 1.51     | 1.52     | 1.49        | 1.49         |
> > > | Quantized SLiM + SLiM-Quant    | 1.32     | 1.39     | 1.43     | 1.50     | 1.51     | 1.52     | 1.49        | 1.49         |
> > >
> > > As illustrated in the above tables, SLiM with quantized adapters (Quantized SLiM + SLiM-Quant) achieves comparable memory and compute reductions to other methods that do not utilize low-rank adapters. This demonstrates that the additional parameter overhead introduced by our low-rank adapters (with $r=0.1$) is minimal, preserving efficiency while providing the benefits of adaptive compression. We have included these statistics in the revised manuscript to ensure a fair and transparent comparison.
> > >
> > > ---
> > >
> > > # Justifying the Use of Significance Methods for Low-Rank Adapter Optimization
> > >
> > >
> > > Thank you for this insightful question. The significance methods, like the one used in Wanda, identify which weights contribute the most to the layer’s output, highlighting the parts of the model that are most impactful for maintaining performance. When introducing low-rank adapters, our goal is to minimize the output recovery error for each layer. By leveraging significance metrics, we can selectively recover the weights that are critical for the layer’s output, rather than simply focusing on the magnitude of the compression errors.
> > >
> > > Optimizing based on significance rather than just error magnitude allows us to better preserve the model’s original performance. As demonstrated in our results, this approach leads to more effective low-rank adaptations, significantly enhancing model accuracy compared to naive methods that do not take weight importance into account.
> > >
> > > ---

---

> > > > ### Author Response · Authors · 2024-11-21
> > > > **Motivation of Equation 8, Wanda’s Additive Property, Low-Rank Adapter Significance, Advanced Joint Quantization, and Computational Overhead**
> > > >
> > > > # Part 4 / 4
> > > >
> > > > # Explaining the Higher Error Rate from Adding Sparsity to LQ-LoRA
> > > >
> > > > Thank you for raising this point. In general, imposing sparsity tends to introduce a higher error compared to 4-bit quantization. This difference is evident when comparing state-of-the-art sparsity techniques (such as Wanda and SparseGPT) with leading quantization methods (like OPTQ, AWQ, and OmniQuant). Sparsity often disrupts the model's weight structure more significantly, which can degrade performance. Therefore, when adding sparsity on top of methods like LQ-LoRA, the resulting error rate increases substantially due to the compounded impact of both low-rank adaptation and weight sparsification. This explains why LQ-LoRA performs better without additional pruning.
> > > >
> > > >
> > > >
> > > > Thank you for your valuable feedback. We agree that understanding the parameter overhead is important for a fair comparison. In practice, the quantization and dequantization of weights are fused with matrix multiplications in Float-16, which minimizes any additional overhead. This approach is consistent with methods like OmniQuant, where the quantization process does not impose significant performance penalties. Therefore, we did not focus on quantization's impact in the Inference Speedup section of the manuscript.
> > > >
> > > > Regarding the theoretical computations of operations (OPs), we acknowledge that these details were not included. As mentioned in the paper, and to the best of our knowledge, there is currently no open-source implementation for Int-4 x Int-8 sparse matrix-matrix multiplication CUDA kernels. Implementing such kernels is beyond the scope of this work but is planned as a part of future work. We appreciate your understanding of these constraints.
> > > >
> > > > ---
> > > >
> > > > Thank you again for your insightful comments. We trust that our responses have clarified your concerns and justify a re-evaluation of the score. We are eager to engage in further discussion and are available to address any remaining questions during the discussion phase. We truly appreciate your time and effort.
> > > >
> > > > ---
> > > >
> > > > [1] Lin, et al. “AWQ: Activation-aware Weight Quantization for On-Device LLM Compression and Acceleration.” MLSys 2024
> > > >
> > > > [2] Frantar, et al. “SparseGPT: Massive language models can be accurately pruned in one-shot.”, ICML 2023
> > > >
> > > > [3] Hassibi, et al. “Optimal brain surgeon and general network pruning.”
> > > >
> > > > [4] Guo, et al. “Compressing large language models by joint sparsification and quantization.” ICML 2024
> > > >
> > > > [5] Shao, et al. “OmniQuant: Omnidirectionally Calibrated Quantization for Large Language Models.” ICLR 2024.
> > > >
> > > > [6] Ma, et al. “AffineQuant: Affine Transformation Quantization for Large Language Models.” ICLR 2024.

---

> ### Comment · Reviewer_6F8M · 2024-12-01
>
> Thanks to the author's response. But I still have concerns about the novelty and hardware practicality of the proposed method, i.e. actual hardware deployment for compressed LLMs rather than just theoretical or unit analysis. I also notice the same concerns from Reviewer 7uwS and i42N. Finally, I decided to keep my score.

---

### Author Response · Authors · 2024-11-21
**Thank you!**

Dear ACs and Reviewers,

Thank you for taking time to review our manuscript and for your valuable feedback. We appreciate the positive feedback of the reviewers, commending SLiM for its (1) highly novel idea and innovative approach for retaining LoRA during inference ([Reviewer 6F8M](https://openreview.net/forum?id=Usa4pF1e5I&noteId=qHg1W7bgX1)), (2) improved accuracy and novel low-rank adapter ([Reviewer 7uwS](https://openreview.net/forum?id=Usa4pF1e5I&noteId=QyIhjnLAxc)), and reduced memory usage and inference time without retraining while maintaining high accuracy in addition to significant reduction in the time and resources needed for an optional fine-tuning ([Reviewer i42N](https://openreview.net/forum?id=Usa4pF1e5I&noteId=kp4dq7YC3O)). We believe that your comments were helpful in further improving our work. As a result, we have incorporated additional clarifications and results as summarized below:

- [Rebuttal-A1](https://openreview.net/forum?id=Usa4pF1e5I&noteId=qHg1W7bgX1), [Rebuttal-A3](https://openreview.net/forum?id=Usa4pF1e5I&noteId=kp4dq7YC3O): Comparing SLiM against multiple one-shot pruning and quantization papers, including JSQ [1], AWQ [2], OmniQuan t[3], and AffineQuant [4].
- [Rebuttal-A1](https://openreview.net/forum?id=Usa4pF1e5I&noteId=qHg1W7bgX1), [Rebuttal-A2](https://openreview.net/forum?id=Usa4pF1e5I&noteId=QyIhjnLAxc), [Rebuttal-A3](https://openreview.net/forum?id=Usa4pF1e5I&noteId=kp4dq7YC3O): Memory and compute reduction analysis for SLiM and other pruning and quantization methods.

We believe that we have addressed all the reviewers’ comments individually. We look forward to a productive discussion during the author response period.

Best regards,

Authors

---

[1] Guo, et al. “Compressing large language models by joint sparsification and quantization.” ICML 2024

[2] Lin, et al. “AWQ: Activation-aware Weight Quantization for On-Device LLM Compression and Acceleration.” MLSys 2024

[3] Shao, et al. “OmniQuant: Omnidirectionally Calibrated Quantization for Large Language Models.” ICLR 2024.

---

### Author Response · Authors · 2024-11-27
**Additional Speedup Experiments for SLiM**

Dear Reviewers,

Thank you for your insightful feedback regarding the speedup analysis of SLiM. We have conducted additional experiments to address concerns about the practical acceleration of our method, and we would like to share these findings with you.

As highlighted in our rebuttal, there is currently no open-source repository supporting the combination of 2:4 sparsity and the specific 4-bit quantization approach employed in our work. This limitation restricts us to analyzing scenarios involving either sparsity or quantization individually, where efficient implementations exist. While implementing optimized kernels for simultaneous sparsity and quantization is beyond the scope of this paper, we believe it is a valuable direction for future work.

To provide a clearer picture of SLiM's potential speedup, we performed an additional experiment to establish a **lower bound** on the achievable acceleration. In this study, we applied 8-bit weight quantization and incorporated low-rank adapters with a rank ratio of 0.1 into each feedforward layer of the model. These experiments were conducted on NVIDIA RTX-3060 GPUs using a batch size of 8, reflecting typical inference-time configurations for large language models.

The table below summarizes the layer-wise speedups observed for various weight sizes in LLaMA and OPT models, with dense models using float-16 weights as the baseline:

### 8-Bit Weight Quantization Speedup ($\times$)

AR denotes the aspect ratio (output dimension / input dimension) of the layer. $\uparrow$ indicates better performance.

| Hidden Dimension       | 768   | 1024  | 2048  | 2560  | 4096  | 5120  | 12288 | 16384 |
|-------------------------|-------|-------|-------|-------|-------|-------|-------|-------|
| Down-Projection (AR=1/4) | 1.39  | 1.6   | 2.43  | 3.35  | 3.25  | 3.45  | 3.1   | 3.27  |
| Up-Projection (AR=4)     | 0.46  | 0.65  | 1.01  | 2.17  | 1.55  | 2.69  | 1.82  | 1.88  |
| Attention Layers (AR=1)  | 0.42  | 0.62  | 1.24  | 2.12  | 1.76  | 2.92  | 2.47  | 2.69  |

We also present a detailed visualization of these results in Figure 2 (Page 18) of the revised manuscript. These findings demonstrate that the inclusion of low-rank adapters does not significantly impede the speedup benefits of quantization, except in specific cases involving small hidden dimensions (e.g., <1024), which correspond to model sizes under 125M parameters. In more practical scenarios involving larger models with billions of parameters, the combination of quantization and low-rank adapters consistently yields substantial speedups.

It is important to note that these results represent a **lower bound** on the speedup achievable with SLiM. Introducing 2:4 sparsity and reducing the quantization bitwidth to 4 bits would further enhance these speedups.

We hope these additional experiments address any remaining questions about the performance implications of our method. Thank you for your valuable input, which has greatly contributed to strengthening this work.

---

### Meta-Review · Area_Chair_RsFH · 2024-12-16

**Metareview:**

This paper presents a post-training method for compressing large language models (LLM). It merges quantization and pruning to reduce the computational overhead, optionally followed by a fine-tuning stage to improve the model's performance. The reviewers are generally negative on this paper, with final scores of 3/3/5, making this case a clear rejection. The reviewers are unanimously concerned about the novelty, technical motivation, and running performance of the proposed method, and the authors did not fully address these concerns during the rebuttal. The AC finds no reason to overturn the reviewers' recommendation.

**Additional Comments On Reviewer Discussion:**

Reviewers acknowledged that they read the rebuttal and decided to keep the original (negative) scores. The rebuttal did not fully address the reviewers' concerns.

---

### Decision · Program_Chairs · 2025-01-22

Reject